

# All-Sky Direct Aerosol Radiative Effects Estimated from Integrated A-Train Satellite Measurements

Meloë S.F. Kacenelenbogen[1], Ralph Kuehn[2], Nandana Amarasinghe[3], Kerry Meyer[1], Edward Nowottnick[1], Mark Vaughan[4], Hong Chen[5], Sebastian Schmidt[5], Richard Ferrare[4], John Hair[4], Robert Levy[1], Hongbin Yu[1], Paquita Zuidema[6], Robert Holz[2], Willem Marais[2]

[1]NASA Goddard Space Flight Center, Greenbelt, MD, USA

[2]Cooperative Institute for Meteorological Satellite Studies, Space Science and Engineering Center, University of Wisconsin—Madison, Madison, Wisconsin, USA

[3]Science Systems and Applications Inc/NASA Goddard Space Flight Center, Greenbelt, MD, USA

[4]NASA Langley Research Center, Hampton, Virginia, USA

[5]Department of Atmospheric and Oceanic Sciences, University of Colorado, Boulder, CO, USA

[6]Rosenstiel School of Marine and Atmospheric Sciences, University of Miami, Miami, Florida, USA

*Correspondence to*: Meloë S. F. Kacenelenbogen (meloe.s.kacenelenbogen@nasa.gov)

**Abstract.** Improved satellite-derived observations of the Direct Aerosol Radiative Effects (DARE) remain essential to reduce the uncertainty in the impact of aerosol on solar radiation. We develop a framework to compute DARE at the top of the Earth's atmosphere, in the short-wave part of the electromagnetic spectrum and in all-sky conditions along the track of the A-Train constellation of satellites. We use combined state-of-the-art aerosol and cloud properties from satellite sensors Cloud-Aerosol Lidar with Orthogonal Polarization (CALIOP) and Moderate Resolution Imaging Spectroradiometer (MODIS). We also use a global reanalysis from the Modern-Era Retrospective analysis for Research and Applications Version 2 (MERRA-2) to provide vertical distribution of aerosol properties and atmospheric conditions. Diurnal mean satellite DARE values range from -25 (cooling) to 40 W·m$^{-2}$ (warming) over the Southeast Atlantic during three days from the NASA ObseRvations of Aerosols above CLouds and their intEractionS (ORACLES) aircraft campaign. These three days also show agreement between our satellite DARE and co-located airborne Solar Spectral Flux Radiometer (SSFR) measurements. This paper constitutes the first step before applying our algorithm to many more years of combined satellite and model data over many regions of the world. The goal is to ultimately assess the order of importance of atmospheric parameters in the calculation of DARE for specific aerosol and cloud regimes. This will inform future missions where, when and how accurately the retrievals should be performed to reduce all-sky DARE uncertainties.

**Key Points.**

● Our semi-observational estimates of all-sky Direct Aerosol Radiative Effect (DARE) along the orbital track compare well with suborbital measurements during the ORACLES field campaign over the Southeast Atlantic.



36●     This paper constitutes the foundation for extending the algorithm to broader regions and multiple years to assess the

37     order of importance of atmospheric parameters in the calculation of DARE for specific aerosol and cloud regimes.

38●     We discuss the limitations in our semi-observational satellite all-sky DARE results

## 1 Introduction

Small suspended individual particles (aerosols) can either scatter, reflect or absorb incoming sunlight (also called aerosol-radiation interactions) and influence cloud properties (also called aerosol-cloud interactions), both perturbing the radiation balance of the Earth-atmosphere system. The total radiative effects resulting from aerosol-radiation and aerosol-cloud interactions play a key role in the Earth's climate as they offset roughly one-third of the warming from anthropogenic greenhouse gases (Foster et al., 2021). Reducing uncertainties in the total aerosol radiative effects largely contributes to reducing uncertainty in quantifying present-day climate change (Foster et al., 2021). Although uncertainties in aerosol-cloud interactions dominate the total aerosol radiative forcing (with a global anthropogenic aerosol radiative forcing of -1.0 ± 0.7 W·m$^{-2}$), uncertainties due to aerosol-radiation interactions are still on the order of 100% (with a global anthropogenic radiative forcing of -0.3 ± 0.3 W·m$^{-2}$) (Foster et al., 2021). Note that these uncertainties represent model diversity and are generally a lower bound on uncertainty (e.g., Li et al., 2022). To illustrate, Myhre et al. (2013) conducted aerosol comparisons between observations and models, and reported a large inter-model spread in the Radiative Forcing due to aerosol-radiation interactions (RFari) of the aerosol species. For example, this is illustrated by a range from 0.05 to 0.37 W·m$^{-2}$ in RFari of Black Carbon (BC, the dominant light absorbing biomass burning (BB) smoke aerosol component across all visible wavelengths), and a standard deviation of 0.07 W·m$^{-2}$ compared to a mean RFari of 0.18 W·m$^{-2}$ of BC (i.e., a 40% relative standard deviation). Our study focuses on aerosol-radiation interactions in the shortwave (SW) part of the electromagnetic spectrum (i.e., four broad band channels between 345 nm and 1242 nm to be exact), at the Top-Of-Atmosphere (TOA), in all-sky conditions (i.e., in clear and cloudy skies) without distinguishing between aerosols from human-made (anthropogenic) or natural sources, and without consideration of pre-industrial times from a climatological perspective.

The TOA SW Direct Aerosol Radiative Effects (DARE) – referred to as DARE in W·m$^{-2}$ – quantifies the change in the net radiative flux at TOA, F$^{net}$, due to perturbations in the loading of aerosol in the atmosphere, which can be expressed by the following equation:

$$
\begin{aligned}
DARE_{TOA} &= F^{net}_{aerosol\ present} - F^{net}_{no\ aerosol\ present} \\
&= \left( F^{\downarrow,TOA}_{aerosol\ present} - F^{\uparrow,TOA}_{aerosol\ present} \right) - \left( F^{\downarrow,TOA}_{no\ aerosol\ present} - F^{\uparrow,TOA}_{no\ aerosol\ present} \right)
\end{aligned} \tag{1}
$$

where $F^{\downarrow}$ and $F^{\uparrow}$ are the downwelling and upwelling flux. Since the incoming solar radiation is the same (i.e., $F^{\downarrow,TOA}_{aerosol\ present} = F^{\downarrow,TOA}_{no\ aerosol\ present}$), DARE can be simplified as the change in the upwelling radiative flux at TOA (i.e., $F^{\uparrow,TOA}_{no\ aerosol\ present} - F^{\uparrow,TOA}_{aerosol\ present}$).





A negative DARE indicates a cooling effect because more energy leaves the Earth's climate system, while a positive
DARE indicates a trap of energy in the climate system or a warming effect. The magnitude and sign of DARE depends
on extensive aerosol properties (which are associated with aerosol loading), intensive aerosol properties (which are
associated solely with aerosol type) and the reflectivity of the underlying surface (e.g., Yu et al., 2006; Chand et al.,
2009; Wilcox et al., 2012; Peters et al., 2011; De Graaf et al., 2012, 2014; Meyer et al., 2013, 2015; Peers et al., 2015;
Feng and Christopher, 2015). For example, even for a homogeneous aerosol layer, Russell et al. (2002) showed how
DARE can switch from negative values (cooling) in clear skies over oceans (low surface albedo) to positive values
(warming) over clouds (high surface albedo).

Substantial progress has been made in the estimation of DARE in clear skies using satellite observations (e.g., Yu et
al., 2006; Oikawa et al., 2013, 2018, Matus et al., 2015, 2019, Korras-Carraca et al., 2019, Lacagnina et al., 2017,
Thorsen et al., 2021). However, fewer studies use satellite observations to estimate DARE above thick clouds, and
even fewer studies are devoted to DARE estimates above all types of clouds (e.g., De Graaf et al., 2012, 2014; Meyer
et al., 2013, 2015; Zhang et al., 2016; Thorsen et al., 2021). The number of studies examining DARE below thin
clouds is vanishingly small. By not including aerosols below thin clouds in all-sky DARE calculations, a significant
portion of the total aerosol effect on radiation is missed. Previous studies listed in Thorsen et al. (2021) show a wide
range of DARE values using satellites, i.e., from -3.1 to -0.61 W·m⁻² in all-skies and from -7.3 to -2.2 W·m⁻² in clear-
skies. This is why we need to further reduce the overall (still significant) uncertainties in observational DARE. As
such, it is important to account for the vertical order, location and amount of different tropospheric aerosol types, the
ocean and cloud reflectivity using satellite observations to calculate DARE.

In this paper, we develop a framework to compute a semi-observational DARE along the track of the A-Train
constellation of satellites using combined aerosol and cloud properties from state-of-the-art satellite sensors
CALIOP/CALIPSO and MODIS/Aqua. MERRA-2, a global reanalysis that assimilates space-based observations of
aerosols is used to provide additional aerosol intensive properties and atmospheric conditions. We use MODIS-derived
pixel-level cloud properties such as Cloud Fraction (CF) and the cloud albedo, which is mostly informed by the Cloud
Optical Thickness (COT), and the Cloud droplet Effective Radius (CER) (note that Cloud Water Path (CWP) can also
be derived from COT and CER) (Twomey, 1974). CF is the percentage of a given pixel in a satellite image that is
covered by clouds. COT is a measurement of how much light is scattered and reflected by clouds, indicating how
"thick" clouds appear to be. CER represents the average size of cloud droplets. CWP is a measurement of the total
amount of liquid water contained within a vertical column of a cloud, indicating how much water is present in clouds.

CALIOP and MERRA-2 aerosol properties used in all-sky DARE calculations are the spectral Aerosol Optical Depth
(AOD), Single Scattering Albedo (SSA) and asymmetry parameter (ASY), as well as the aerosol vertical distribution
in the atmosphere, and particularly its location relative to clouds. AOD is a measure of the extinction of sunlight due
to aerosols that depends on the aerosol amount and aerosol type (e.g., for a fixed loading and relative humidity, the
AOD of smoke will be significantly higher than the AOD of marine aerosols). SSA is a measure of aerosol light



scattering over light extinction which depends on the light absorption (i.e. the aerosol composition) and the aerosol
size. ASY is a measure of the directionality of scattered light from the aerosol (e.g., if the radiation is scattered back
to space, there is a loss of energy for the Earth's climate system) and depends on particle shape. The spectral
dependence of the AOD is a first-order indication of the effective size of the aerosol particles. To illustrate the effective
particle size of the aerosol (to the first order) in our study, we introduce the Extinction Angstrom Exponent (EAE)
parameter, the ratio of two aerosol extinction coefficients at two different wavelengths divided by the ratio of these
two wavelengths in log space. Coarse size mode-dominated particles (e.g., dust aerosols) usually record smaller EAE
values compared to fine-mode dominated particles (e.g., smoke). Finally, the spectral shape of SSA is useful for
distinguishing between different types of absorbing aerosols (e.g., Russell et al., 2014; Kacenelenbogen et al., 2022).

We compute DARE for three specific days over the Southeast Atlantic (this paper) as a first step before we extend our
study to multiple years and other regions of the globe (follow up paper(s)). We carefully select our case studies such
that our semi-observational satellite DARE results can be validated against airborne observations from the ORACLES
campaign. Several studies have attempted to estimate DARE over the Southeast Atlantic (see, for example, the studies
listed in Table 1 in Kacenelenbogen et al., (2019)). This region is known to show global maximum positive DARE
values (e.g., Waquet et al., 2013). According to Jouan et al. (2024), the long-term increase of biomass burning aerosols
over the Southeast Atlantic could represent an underrecognized source of global warning (i.e., all-sky DARE has
become more positive, $+0.04 \pm 0.15\,\mathrm{W\,m^{-2}\,yr^{-1}}$, due to aerosols in cloudy sky regions). Note that the long-term increase
of smoke over this region can be attributed to increased warm temperature advection and strengthening of the easterly
winds over time (Tatro and Zuidema, 2025)

The paper is organized as follows - Section 2 describes a framework to compute DARE in the case of a few identified
atmospheric scenarios along the satellite track. Section 3 presents our semi-observational estimates of DARE, the
inputs of aerosol and cloud parameters, and comparisons against field campaign measurements during our three case
studies. Sections 4 and 5 discuss future work and conclude our paper.

**2 Data and Method**
In this paper, we present two sets of DARE. First, a DARE$_S$ parameter that uses observations from satellite sensors
and estimations from a model (see section 2.1) and represents the main results of our study. Second, a parametrized
DARE$_P$ parameter based on Cochrane et al. (2021) and used as one of two ways to evaluate our DARE$_S$ results (see
section 2.2). Table 1 defines the acronyms used to describe the satellite-derived and model-based computational inputs
to the DARE calculations. Table 2 summarizes the steps required to calculate estimates of DARE$_S$ and DARE$_P$. The
subsections of section 2 describe the contents of Table 2 in further detail.




| | Input Parameter to DARE Calculation | Description |
|---|---|---|
| CALIOP (~1/3 km) | CALIOP$_{ACAOD\_standard}$ or CALIOP$_{AOD\_standard}$ | CALIOP above-cloud AOD (ACAOD) or total column AOD at 532nm obtained by integrating the standard CALIOP version 4.51 (V4.51) aerosol extinction profile (Young and Vaughan, 2009) between the aerosol top and base heights above clouds or in clear skies |
| | CALIOP$_{ACAOD\_DR}$ | CALIOP V4.51 above-cloud AOD at 532 nm derived using the depolarization ratio (DR) method described in Hu et al. (2007) |
| | CALIOP$_{ODAOD}$ | CALIOP V4.51 total column AOD at 532 nm estimated using the Ocean Derived Aerosol Optical Depths (ODAOD) product (Ryan et al., 2024) |
| | CALIOP$_{vfm}$ | CALIOP V4.51 Vertical Feature Mask (VFM) reports detected layer heights and identifies aerosols and clouds according to type and subtype (Vaughan et al., 2009; Liu et al., 2010) |
| | MODIS$_{Cloud}$ (1km) | (i) Cloud optical/microphysical properties and cloud-top property retrievals from MODIS/VIIRS CLDPROP Version-1.1 (Platnick et al., 2021)<br>(ii) MODIS aerosol and cloud products corrected for overlying aerosols using a new aerosol radiative model (Meyer et al., 2015) |
| | MERRA-2 (~55 km) | Atmospheric composition and weather profiles from the Modern-Era Retrospective analysis for Research and Applications, Version 2 (Gelaro et al., 2017) |

**Table 1: Acronyms used to describe computational inputs to DARE$_S$ and DARE$_P$ calculations in Table 2.**

| | $\mathbf{DARE_S}$ | $\mathbf{DARE_P}$ |
|---|---|---|
| **Atmospheric Scenarios** | Aerosol above and below a single low level (<3km) thick, thin and/ or broken liquid cloud and aerosol in (mostly) clear skies (§ 2.1.3) | |
| **Model** | RRTMG-SW | Eq. 12 of Cochrane et al. (2021) |
| **Cloud Detection and Characterization** | **CALIOP**$_{vfm}$ and **MODIS**$_{Cloud}$ to select qualifying clouds and to define thick, broken and/ or thin clouds in atmospheric scenarios; **MODIS**$_{Cloud}$ to assign cloud properties (i.e., CWP, CER, COT) (§ 2.1.1) | |
| **Cloud Albedo** | N/A | Computed for RRTMG bands (25) using Mie calculations and DISORT |
| **Cloud Top Height (CTH) and Cloud Base Height (CBH)** | **CALIOP**$_{vfm}$ for CTH; CBH = CTH - 500m | N/A |
| **Uppermost Aerosol Top Height (ATH) and lowermost Aerosol Base Height (ABH)** | **CALIOP**$_{vfm}$ for ATH above clouds and ATH and ABH in clear skies; ABH = CTH above clouds; **MERRA-2** for ATH and ABH below clouds | N/A |
| **Vertical distribution of spectral ASY and SSA** | MERRA-2 (§ 2.1.1) | N/A |
| **Vertical distribution of spectral aerosol extinction coefficient** | Below clouds, we use **MERRA-2**; elsewhere (above clouds and clear-sky), **MERRA-2** normalized spectral aerosol extinction coefficient is multiplied by **CALIOP** AOD at 532nm (i.e., a combination of CALIOP$_{ACAOD_standard}$, CALIOP$_{ACAOD_DR}$, CALIOP$_{AOD_standard}$, and CALIOP$_{ODAOD}$) (§ 2.1.2, Fig. 1) | |
| **Diurnal cycle of aerosols and clouds** | We only vary SZAs during the day, assuming constant aerosol and cloud properties | |
| **Atmospheric composition, weather and ocean surface winds**[(1)] | **MERRA-2** (§ 2.1.1) | N/A |
| **Ocean Surface BRDF** | Cox-Munk BRDF [Jin et al., 2011] with Chlorophyl concentration = 0.2 g/m$^3$ | Standard Lambertian with an albedo value of 0.03 |
| **ΔDARE calculation** | Compute upper and lower bounds using uncertainties listed in Table 4 | N/A |

**Table 2: Two different DARE calculations (i.e., semi-observational $DARE_S$, and parametrized $DARE_P$) in our study and their respective inputs. RRTMG-SW stands for Short-wave Rapid Radiative Transfer Model. See Table 1 for a description of CALIOP$_{ACAOD_standard}$, CALIOP$_{ACAOD_DR}$, CALIOP$_{AOD_standard}$, CALIOP$_{ODAOD}$, CALIOP$_{vfm}$, MODIS$_{Cloud}$ and MERRA-2. (1) These parameters are assumed constant along the satellite track: $CO_2$ volume mixing ratio = 400 ppmv, $N_2O$ mass density = 0.3 ppmv, $CH_4$ mass density = 1.7 ppmv, $O_{2\ mass\ density}$ = 0.0 kg m$^3$; the ocean surface wind values vary along the satellite**



**track and are provided by MERRA-2; the profiles of temperature, pressure, air density (calculated from pressure and**
**temperature), water vapor and $O_3$ vary along the satellite track and are also provided by MERRA-2.**

To estimate $DARE_S$ in section 2.1, we perform solar broadband radiative transfer (RT) calculations using the
Shortwave Rapid Radiative Transfer Model for General Circulation Model (GCM) applications (RRTMG-SW) RT
code (hereafter, only called RRTMG) (Clough et al., 2005; Iacono et al., 2008) (see Table 2). In RRTMG, gaseous
absorption is treated using the correlated-k approach (Mlawer et al., 1997); the delta-Eddington (Joseph et al., 1976)
two-stream approximation (Meador and Weaver, 1980; Oreopoulos and Barker, 1999) is used for scattering
calculations. Therefore, RRTMG does not need information on the aerosol phase function, which is why we only use
ASY as input. Broadband solar fluxes are calculated from 14 broadbands with bandwidths ranging from 0.2 to
12.0 μm. The four SW RRTMG broadband channels are between 345-442, 442-625, 625-778 and 778-1242 nm. As
listed in Table 2, inputs for RRTMG include the optical properties of aerosol and cloud, atmospheric profiles, ocean
surface BRDF and Solar Zenith Angle (SZA) information. In RRTMG (using two-stream approximation), total fluxes
have an accuracy within 1-2 W·m$^{-2}$ relative to the standard RRTM-SW (using DISORT) in clear sky and in the
presence of aerosols and within 6 W·m$^{-2}$ in cloudy sky. RRTM-SW with DISORT itself is accurate to within 2 W·m$^{-2}$
$^2$ of the data-validated multiple scattering model, CHARTS (https://github.com/AER-RC/RRTMG_SW) (Iacono et
al., 2008).

In this study, we compute both the instantaneous DARE along the satellite track for a given location and time and an
estimated diurnal average DARE at the same location that accounts only for the varying solar angle throughout the
day. In other words, the instantaneous DARE uses SZA at a given CALIOP-derived latitude, longitude and date. We
then vary SZA corresponding to every hour at the same location and date, compute DARE and average all
instantaneous DARE to obtain diurnal mean (or 24h) DARE.

**2.1 Semi-Observational $DARE_S$ Calculations**
To design the algorithm that computes semi-observation-based $DARE_S$ results and to gain understanding of DARE
sensitivities from idealized cases, we first compute a theoretical-based cloudy DARE parameter (that we call $DARE_T$)
using RT calculations on several canonical atmospheric cases. Like Table 2 for $DARE_S$ and $DARE_P$, Table A1 in the
appendix lists the input parameters to our $DARE_T$ calculations -- $DARE_T$ is computed for two types of single low
warm liquid clouds (i.e., COT=1, CER=12 and CWP=8 vs. COT=10, CER=12 and CWP=80) and varying vertical
distributions of RRTMG "build-in" aerosol types (see Fig. A1) while keeping cloud heights, AOD, ASY, atmospheric
composition, weather and ocean surface BRDF constant. We compute $DARE_T$ for thirty-two canonical cases
(illustrated in panel (a-d) of Fig. A2) where we vary the order and amount of two aerosol types over clouds in the
vertical. No matter which type and which vertical distribution of aerosol above cloud is considered, $DARE_T$ values
are lower when aerosols are present above a cloud of COT equal to 1 (case (b) and (d)), compared to a COT equal to
10 (case (a) and (c) in Fig. A2) -- see respectively ~-7 to ~-1 W·m$^{-2}$ for (b-d) vs. ~9 to ~24 W·m$^{-2}$ for (a-c) in the
bottom panel (e) of Fig. A2. We also record lower $DARE_T$ values when adding more scattering aerosols (i.e.,
"continental" aerosol type) to already absorbing aerosols (i.e., "urban" aerosol type). $DARE_T$ values drop from ~24 to





~14 W·m$^{-2}$ when aerosols are present above a cloud of COT equal 10 (see C1-C4 in (a) vs. C5-C8 in (a) in the bottom
panel (e) of Fig. A2) and drop from ~-1 to ~-5 W·m$^{-2}$ when aerosols are present above a cloud of COT equal 1 (see
C1-C4 in (b) vs. C5-C8 in (b) in the bottom panel (e) of Fig. A2). In conclusion, the variability of these DARE$_T$
calculations confirm, as expected, that our semi-observational DARE$_S$ calculations need to account for the vertical
order and location of aerosol types and aerosol amount.

As listed in Table 2, DARE$_S$ uses a mix of satellite and model products as input parameters to RRTMG. Section 2.1.1
describes these satellite and model products in further detail. Section 2.1.2 provides more information on how these
products are combined. Section 2.1.3 describes how we divide the atmosphere into four atmospheric scenarios along
the satellite track. Section 2.1.4 describes the DARE$_S$ uncertainty calculations.

**2.1.1 Data**
**CALIOP/ CALIPSO** flew onboard the CALIPSO platform for 17 years from 2006 to 2023. From launch in April
2006 until September 2018, CALIPSO flew in tandem with multiple other platforms as part of the A-Train
constellation of Earth-observing satellites. CALIOP measured high-resolution vertical profiles of attenuated
backscatter (at 532 nm and 1064 nm) and volume depolarization ratios (at 532 nm) from aerosols and clouds in the
Earth's atmosphere from the surface up to ~40 km. Full instrument details are given in Hunt et al. (2009). A succession
of sophisticated retrieval algorithms is used to derive CALIOP Level 2 products from the Level 1 products (Winker
et al., 2009). These retrieval algorithms are composed of a feature detection scheme (Vaughan et al., 2009), a module
that first distinguishes cloud from aerosol (Liu et al., 2019) and then partitions clouds according to thermodynamic
phase (Avery et al., 2020) and aerosols according to subtypes (Kim et al., 2018; Tackett et al., 2023), and, finally, an
extinction algorithm (Young et al., 2018) that retrieves profiles of aerosol backscatter and extinction coefficients and
the total column AOD based on modeled values of the extinction-to-backscatter ratio (also called lidar ratio) inferred
for each detected aerosol layer subtype.

Previous studies have shown that CALIOP standard AOD products underestimate AOD in clear skies
(Kacenelenbogen et al., 2011; Thorsen et al., 2017; Toth et al., 2018) and above clouds (e.g., Kacenelenbogen et al.,
2014, Rajapakshe et al., 2017), mostly because CALIOP does not detect tenuous aerosol layers having attenuated
backscatter coefficients less than the CALIOP detection threshold (Rogers et al., 2014). The low biases in the total
column and above cloud AODs, denoted in this work as, respectively, CALIOP$_{AOD\_standard}$ and CALIOP$_{ACAOD\_standard}$
(see Table 1; AC stands for Above Cloud), motivates us to also use two new, independently derived estimates of
column optical depth at 532 nm. The first of these uses the depolarization ratio (DR) method developed in Hu et al.
(2007), hereafter called CALIOP$_{ACAOD\_DR}$ at 532 nm, to calculate total column optical depths above opaque water
clouds. By leveraging the unique relationship between layer-integrated volume depolarization ($\delta_v$) and the layer-
effective multiple scattering factor in opaque liquid water clouds (Hu et al., 2006), together with characteristic values
of water cloud lidar ratios, an accurate estimate of the opaque water cloud integrated attenuated backscatter in clear
skies ($\gamma'_{clear}$) can be obtained (Platt, 1973). The two-way transmittance due to aerosols above the cloud (and hence





above cloud optical depth) is thus obtained by dividing the measured cloud integrated attenuated backscatter, $\gamma'_{measured}$,
by the $\gamma'_{clear}$ estimate. As of CALIOP's version 4.51 data release, $CALIOP_{ACAOD\_DR}$ retrievals are now included as a
standard scientific data set (SDS) contained in the layer products for all averaging resolutions. However, the individual
components required for the DR method (e.g., $\delta_v$ and $\gamma'_{measured}$) were routinely reported in earlier data releases, and
hence AODs derived using the DR method have been used extensively in previous studies (e.g., Chand et al., 2008;
Liu et al., 2015; and Kacenelenbogen et al., 2019). Furthermore, comparisons made by Ferrare et al. (2017) show that
$CALIOP_{ACAOD\_DR}$ agrees well (bias and RMS differences less than 0.05 and 10%) with coincident measurements by
the NASA Langley Research Center airborne High Spectral Resolution Lidars (HSRL) during two flights (18 and 20
of September 2016) of the ORACLES field campaign (Redemann et al., 2021). The second of CALIOP's
independently derived total column AOD estimates is provided by the Ocean Derived Column Optical Depths
(ODCOD) algorithm (Ryan et al., 2024), hereafter called $CALIOP_{ODAOD}$ at 532 nm. As with the DR method estimates,
the ODCOD AOD is a new parameter being reported for the first time in CALIOP's V4.51 data release. ODCOD
works by comparing an idealized parameterization of laboratory measurements of the 532 nm detector impulse
response function (IRF) to space-based measurements of the backscattered energy from the ocean surface. Similar in
operation to the technique employed by Venkata and Reagan (2016), the ODCOD algorithm shifts the IRF model in
time and scales it in magnitude to achieve the best fit to the measured data. When weighted by surface wind speed,
the area under the curve of this shifted and scaled model is directly related to the attenuation of the laser surface return
by the intervening atmosphere. Note that both ODCOD and the DR method report *effective* optical depths; that is, the
product of the true overlying optical depths and a column-effective multiple scattering factor, $\eta_{col}$, where $0 < \eta_{col} \leq 1$.
Because ODCOD AODs are retrieved immediately after executing the CALIOP surface detection algorithm, and prior
to conducting a search for atmospheric layers, no attempt is made to separate multiple scattering and single scattering
contributions made by the overlying particulates (i.e., clouds and/or aerosols). Fortunately, in cloud-free columns
containing only aerosol layers and clear skies, $\eta_{col} \approx 1$ (Young et al., 2018). Consequently, multiple scattering
corrections are neglected in the standard extinction retrieval (Winker et al., 2009) and considered unnecessary in the
ODCOD analyses. The extensive comparisons shown in Ryan et al. (2024) demonstrate that $CALIOP_{ODAOD}$ agrees
well with coincident HSRL measurements during all CALIOP-HSRL co-located flights from 2006 to 2022. The
median difference in the daytime between $CALIOP_{ODAOD}$ and HSRL AOD is -0.037 ± 0.052 (-12 % ± 25%; N=149)
with $CALIOP_{ODAOD}$ lower and a correlation coefficient of 0.775.

In our study, as listed in Table 2, we use CALIOP to characterize aerosol optical depth above clouds
($CALIOP_{ACAOD\_standard}$, $CALIOP_{ACAOD\_DR}$) and in clear skies ($CALIOP_{AOD\_standard}$, and $CALIOP_{ODAOD}$), to establish
aerosol and cloud top heights ($CALIOP_{vfm}$; VFM stands for Vertical Feature Mask), and as the source for the SZAs in
our $DARE_S$ calculations. We also use the latest CALIOP stratospheric aerosol profile product (version 1.00; Kar et
al., 2019) to correct for attenuation by stratospheric aerosols in $CALIOP_{ACAOD\_DR}$ and $CALIOP_{ODAOD}$. To do that, we
compute a zonal climatology of Stratospheric Optical Depth (SOD) from the equal-angle data product, then interpolate
the zonal data to the latitude grid of the CALIPSO granule observations (see Fig. A3 in the appendix). Finally, we
remove the SOD from $CALIOP_{ACAOD\_DR}$ and $CALIOP_{ODAOD}$. Note that while performing our $DARE_T$ calculations, we



confirmed the importance of considering stratospheric aerosols. Adding stratospheric aerosols between 25-30 km with
a typical AOD value of 0.04 in the stratosphere (Kloss et al., 2021) to tropospheric aerosols above clouds leads to an
absolute difference in $DARE_T$ up to 3.7 W·m$^{-2}$. We also use $CALIOP_{vfm}$ to select clouds of interest (i.e., single layer
low warm liquid clouds) and to define thick, broken and/ or thin clouds in our four atmospheric scenarios described
in section 2.1.3.

Like other papers (e.g., Su et al., 2013), and because satellites are not yet well suited to broadly observe the vertical
profile of aerosol intensive properties, we use a model to complement satellite observations. We have decided to use
MERRA-2 to inform on the vertical distribution of spectral aerosol intensive properties (see Table 2). We use CALIOP
AOD quantities at 532 nm and we populate the 442-625nm RRTMG channel with an observational AOD value. We
then spectrally extrapolate the AOD at 532 nm in the other broadband RRTMG channels of the short-wave part of the
spectrum using the MERRA-2 spectral shape of extinction coefficients as further described in section 2.1.2.

**MODIS/ Aqua** flew as part of the A-Train constellation of satellites from 2002 until a final drag makeup satellite
maneuver in December 2021, after which Aqua began a slow descent below the A-Train. It has 20 shortwave spectral
bands from 412 nm to 2130 nm, along with 16 infrared bands from 3.7 to 14.4μm, enabling retrievals of the
macrophysical, microphysical and radiative properties of clouds. CER commonly is retrieved simultaneously with
COT from passive imager remote sensing observations using a bi-spectral technique (Nakajima and King, 1990;
Platnick et al., 2003) pairing a non-absorbing visible or near-infrared spectral channel sensitive to COT with an
absorbing shortwave infrared or mid-wave infrared spectral channel sensitive to CER. In this paper, we use two types
of cloud products from MODIS (referred to as $MODIS_{Cloud}$ in Table 1). The first type of $MODIS_{Cloud}$ products is from
the current operational algorithm and does not account for the presence of aerosols above clouds (Meyer et al., 2013).
They are called uncorrected $MODIS_{Cloud}$ products in this paper and are derived from the Cross-platform HIgh
resolution Multi-instrument AtmosphEric Retrieval Algorithms (CHIMAERA) shared-core suite of cloud algorithms
(Wind et al., 2020). This suite of algorithms includes cloud optical/microphysical properties (e.g., thermodynamic
phase, optical thickness, particle effective size, water path) and cloud-top property retrievals from MODIS/ Visible
Infrared Imaging Radiometer Suite (VIIRS) CLDPROP Version-1.1, designed to sustain the long-term records of
MODIS (cloud properties continuity product) (Platnick et al., 2021). The second type of $MODIS_{Cloud}$ products derives
from a new retrieval technique that corrects the MODIS cloud retrievals by accounting for overlying aerosols. They
are called corrected $MODIS_{Cloud}$ products in this paper. In Grosvenor et al. (2018), comparisons between MODIS and
Advanced Microwave Scanning Radiometer 2 (AMSR2), which is not sensitive to the above-cloud aerosol, indicate
derived cloud droplet number concentration differences of < 10 cm$^{-3}$ over most of the Southeast Atlantic stratocumulus
deck. As described in Meyer et al. (2015), what we call corrected $MODIS_{Cloud}$ in this paper was achieved by adding
aerosols with prescribed scattering properties in the radiative transfer calculations that are used to construct bi-spectral
lookup tables. Note that this correction is strongly dependent on the assumed aerosol scattering properties. For this
study, these properties are derived from the NASA Spectrometers for Sky-Scanning Sun-Tracking Atmospheric
Research (4STAR) observations obtained during ORACLES 2016. Since the cases we carefully selected for $DARE_S$



also are during the deployment of ORACLES, we can assess the effects of using either corrected or uncorrected
MODIS$_{Cloud}$ properties in DARE$_S$ calculations (see section 2.1.4). The main cloud properties needed in the DARE$_S$
calculations are CWP, CER and COT (see Table 2).

Regardless of the cloud product used, validating retrievals such as COT, CER, and CWP is difficult as there are no
direct measurements of these radiative quantities. Microphysical retrievals can be compared against airborne in situ
cloud probes, and previous investigations have found notable differences, though strong correlation, between the two,
with MODIS-derived CER on average more than 2µm larger than that derived from legacy in situ probes (e.g.,
Nakajima et al., 1991; Platnick and Valero, 1995; Painemal and Zuidema, 2011; Min et al., 2012; King et al., 2013;
Noble and Hudson, 2015; Gupta et al., 2022). Other studies using probes leveraging different observation techniques
(e.g., Witte et al., 2018) have shown no systematic differences in CER. Comparisons against other retrieval techniques,
such as polarimetry, can also inform on CER retrieval quality. Using ORACLES airborne observations, Meyer et al.,
(2025) performed an extensive comparison of spectral imager liquid CER retrievals (from the Enhanced MODIS
Airborne Simulator, eMAS, an airborne proxy instrument of MODIS, and the Research Scanning Polarimeter, RSP)
with those from polarimetry (from RSP) and CER derived from two in situ cloud probes. Agreement between the
imager, polarimetric, and probe-derived CER, was found to be case- and spectral-dependent, and accounting for
above-cloud aerosol absorption in the bi-spectral imager retrievals (equivalent to using corrected MODIS$_{cloud}$) either
has no impact or worsens the agreement depending on the spectral channel used.

NASA's Global Modeling and Assimilation Office (GMAO) **MERRA-2** data became available in September 2015
(Gelaro et al., 2017), covering 1980 - Present. It is based on a version of the GEOS-5 atmospheric data assimilation
system that was frozen in 2008 and was produced on a 0.5 x 0.625º grid (~55 km x 69 km) on 72 hybrid sigma-
pressure coordinate system vertical levels. It was frozen so that the underlying model physics, schemes, and data
assimilation techniques are the same for the duration of the MERRA-2 reanalysis. It uses a version of the Goddard
Chemistry Aerosol Radiation and Transport (GOCART) model (Chin et al., 2002, Colarco et al., 2010, Colarco et al.,
2014) to treat the emission, transport, removal, and chemistry of dust, seasalt, sulfate, and carbonaceous aerosols.
Aerosol optical properties are computed from the Mie-theory based Optical Properties of Aerosol and Cloud (OPAC)
dataset (Hess et al., 1998), except for dust, which were derived by an observation-derived dataset of refractive indices
and an assumption of a spheroidal shape as described in Colarco et al. (2014). MERRA-2 assimilates satellite, air, and
ground observations (Randles et al., 2017) to constrain both the atmospheric and aerosol state in the model. MERRA-
2 also provides optical properties within the SW RRTMG broadband channels. Many papers have shown that
MERRA-2 aerosol extensive, and intensive properties and horizontal/vertical distribution are far from perfect (e.g.,
Nowottnick et al., 2015). For example, GEOS aerosol Single Scattering Albedo (SSA) was shown to be consistently
higher than *in-situ* measurements during the ORACLES field campaign, explained by an underestimation of black
carbon content by the GEOS model (Das et al., 2024). However, since the modeled data provides spatially- and
temporally resolved atmospheric variables assimilated from observations, they can be used as complimentary products
(in addition to satellite products) to inform the calculation of DARE. As listed in Table 2, our DARE$_S$ calculations use





MERRA-2 (GMAO, 2015) ocean surface winds, ozone, temperature, pressure, air density, and water vapor profiles,
and aerosol intensive properties (i.e., spectral extinction coefficient, SSA, ASY) above and below low opaque water
clouds and in clear skies. As CALIOP cannot reliably provide any aerosol information below clouds due to signal
attenuation, we also use MERRA-2 to inform on aerosol extensive, intensive properties and layer heights below
clouds.

Finally, we must assume a consistent observed and modeled extinction coefficient threshold under which we consider
there is no aerosol present in the atmosphere. Based on Rogers et al. (2014), we consider that there are no aerosols
(and hence DARE$_S$ = 0) if the CALIOP extinction coefficient at 532 nm is below 0.07 km$^{-1}$. As the lower threshold
on the CALIOP extinction of 0.07 km$^{-1}$ is based on an aerosol layer that is 1.5 km thick in Rogers et al. (2014), we
impose a lower threshold on MERRA-2 extinction of 0.014 km$^{-1}$ in the 442-625 nm RRTMG broadband channel. This
is because the MERRA-2 layers are, on average, 0.29 km thick in September 2016 in a MERRA-2 grid box located at
[-10°W-10°E, -40°S-0°] and between 1-5 km altitude.

**2.1.2 Combination of Satellites and Model**
In this subsection, we show how we combine the satellite products with modeled data to perform RT calculations that
represents DARE. First, we collocate MODIS and CALIOP satellite observations every 1 km horizontally along
CALIOP's track using the method described in Nagle and Holz (2009). By doing this we account for the parallax
effect, i.e., the cloud top height dependence on spatial colocation. Using a simple surface collocation method that does
not account for the parallax effect could result in a horizontal shift of more than 5 pixels (Holz et al., 2008).

Second, to compute DARE$_S$, we need to combine aerosol extensive properties primarily informed by CALIOP with
aerosol intensive properties primarily informed by MERRA-2. Figure 1 illustrates the combination of CALIOP and
MERRA-2 products above clouds and in clear skies. In the green region in Fig. 1a, we assume one or multiple aerosol
layer(s) of different aerosol types contained between the uppermost CALIOP-informed aerosol top and the lowermost
CALIOP-informed aerosol base heights. The AOD at 532 nm corresponding to the vertical integration of the extinction
coefficients of these single or multiple aerosol layers is informed by CALIOP (called AOD$_C$) on Fig. 1 (i.e., either
CALIOP$_{ACAOD\_standard}$, CALIOP$_{ACAOD\_DR}$, CALIOP$_{AOD\_standard}$ or CALIOP$_{ODAOD}$ -- see Table 1). The illustrative
MERRA-2 profile in Fig. 1b collocated in space and time with the profile in Fig. 1a shows three aerosol layers in blue,
orange and yellow on an initial (and uneven) MERRA-2 vertical grid. It also shows six "aerosol-free" MERRA-2
aerosol layers in hashed grey (i.e., aerosol layers for which MERRA-2 extinction coefficients are below 0.014 km$^{-1}$ in
the 442-625 nm RRTMG broadband channel). We call L, the number of MERRA-2 aerosol layers (L=3 in Fig. 1b). In
each vertical layer, i, and for each RRTMG broadband channel, $\lambda$, we record the MERRA-2 extinction coefficient,
$\sigma_M(i,\lambda)$, MERRA-2 SSA, $SSA_M(i,\lambda)$, and MERRA-2 ASY, $ASY_M(i,\lambda)$ in Fig. 1b. To combine CALIOP and
MERRA-2 (Fig. 1c), we keep L constant and do not allow "aerosol-free" layers (hashed grey) to physically touch
either the CALIOP-inferred aerosol top and aerosol base heights. Combined CALIOP and MERRA-2 (C-M)



$SSA_{C-M}(i,\lambda)$ and $ASY_{C-M}(i,\lambda)$ are directly equal to MERRA-2 $SSA_M(i,\lambda)$, and $ASY_M(i,\lambda)$. The combined CALIOP
and MERRA-2 extinction coefficients, $\sigma_{C-M}(i,\lambda)$, is computed as in Eq. 2:

$$\sigma_{C-M}(i,\lambda) = \left(\frac{AOD_C(532nm)}{L}\right) \times \left(\frac{\sigma_M(i,\lambda)}{\sigma_M(i,442-625nm)}\right) \tag{2}$$



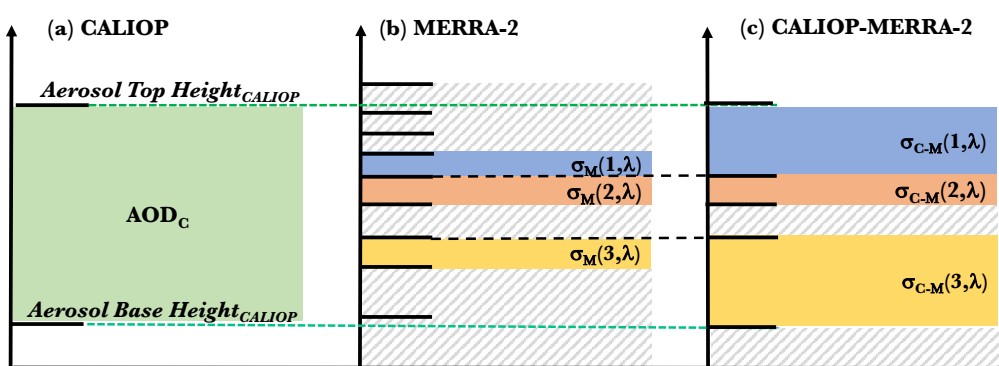



**Figure 1: Illustration of how we combine MERRA-2 and CALIOP above clouds and in clear skies. In green in (a), we**
**assume one or multiple aerosol layers contained between a CALIOP-inferred aerosol uppermost layer height and aerosol**
**lowermost base height. The AOD of the aerosol plume in green is informed by CALIOP at 532 nm, $AOD_C$. (b) is the**
**MERRA-2 profile collocated in time and space to the CALIOP profile in (a). The MERRA-2 profile in (b) shows three**
**aerosol layers (blue, orange and yellow) for which the MERRA-2 extinction coefficient, SSA, and ASY is respectively called**
**$\sigma_M(i,\lambda)$, $SSA_M(i,\lambda)$, and $ASY_M(i,\lambda)$, in each layer i and in each broadband RRTMG channel $\lambda$. The combined CALIOP**
**and MERRA-2 profile in (c) records $SSA_M(i,\lambda)$, $ASY_M(i,\lambda)$, and $\sigma_{C-M}(i,\lambda)$ as computed using Eq. 2.**

**2.1.3 DARES for Four Atmospheric Scenarios**
Based on our theoretical calculations (see Fig. A2) and previous studies such as Matus et al. (2019), DARE results are
clearly dependent on cloud thickness and spatial homogeneity. To evaluate DARE, we generalize the atmospheric
conditions into four scenarios based on different cloud conditions. In assembling our combined CALIPSO + MODIS
data set, we start by removing records that report clouds of any types and at any altitudes above the single low warm



liquid cloud (LWLC) that is closest to the Earth's surface (i.e., above a cloud top height of 3 km). The first three
scenarios show aerosol above and below different types of LWLC and the fourth scenario shows aerosol in (possibly
cloud contaminated) clear skies. We define the geometrical thickness and spatial uniformity of LWLC using both
CALIOP and MODIS cloud properties. Table 3 defines how we call different types of LWLC moving forward, how
they are overall characterized and in which scenario they can be present.






| Atmospheric Scenario | | S1 | S2 | S3 | S4 |
|---|---|---|---|---|---|
| Single Low Warm Liquid Cloud (LWLC) | | Thick and uniform | Thick and broken | Thin, and possibly broken | Small or not present |
| Cloud Conditions | CALIOP N cloud layer @ 1km | 1 | | | <=1 |
| | CALIOP Cloud classification and cloud phase identification (CAD & phase) @ 1km | Highly confident | | | N/A |
| | CALIOP N single shot cloud detected within1km | 3 | >=2 | | <=2 |
| | CALIOP N single shot opaque flag within1km | 3 | 2 | <=1 | <=2 |
| | CALIOP consecutive single shot of non-opaque clouds within1km | FALSE | | TRUE | FALSE |
| | CALIOP consecutive single shot of opaque clouds within1km | TRUE | | FALSE | FALSE |
| | MODIS MOD35 cloud mask @ 1km Bits 1-2 "Unobstructed FOV confidence flag" | N/A | | | 2 or 3 (i.e., "probably clear" or "clear") |
| | MODIS CLDPROP (CLDPROP_L2_MODIS_Aqua) and CLDPROPOACAERO COT @ 1km | > 4 | | < 4 | N/A |
| **Synonyms and Main Features:** | | | | | |
| Thick | Opaque, non-transparent according to CALIOP and COT>4 according to MODIS | | | | |
| Thin | Non-opaque, semi-transparent, or transparent according to CALIOP and COT<4 according to MODIS | | | | |
| Uniform | Non-broken, homogeneous according to CALIOP (i.e., CALIOP detects three consecutive single shot clouds within a 1 km stretch) | | | | |
| Broken | Non-uniform, non-homogeneous according to CALIOP (i.e., CALIOP does not detect three consecutive single shot clouds within a 1 km stretch) | | | | |

**Table 3: Method to distinguish single Low Warm Liquid Cloud (LWLC) in atmospheric scenario S1, S2, S3 and S4 using**
**MODIS and CALIOP. For all scenarios, we collocate MODIS and CALIOP every 1km along the CALIOP track using the**
**method described in Nagle and Holz [2009]; profiles are deleted if high clouds are present with CTH > 3km; clouds are**





**"highly confident" when 111>Cloud-Aerosol Discrimination (CAD)>20; cloud temperature>-10°C; cloud altitude<3km;**
**phase Quality Assurance (QA) >=2; cloud phase ==2.**

Table A3 in the appendix describes which aerosol (CALIOP and/ or MERRA) and cloud (MODIS) parameter was
used and how it was filtered to compute DARE$_S$ in the case of S1, S2, S3 and S4.

We compute DARE$_S$ using the input parameters in Table 2 and for each 1 km stretch to which is attributed a particular
atmospheric scenario in Table 3. We then regroup all these DARE$_S$ results along the track to obtain either daily, and
eventually regional, monthly, seasonal and/ or yearly DARE$_S$ statistics.
When clouds are present in the atmosphere, DARE$_S$ of aerosol above clouds (i.e., DARE$_{cloudy}$ for S1, S2, and S3
combined) is the subtraction of upward fluxes for clouds without aerosols and for clouds with aerosols above them. If
we have N1 x S1, N2 x S2 and N3 x S3 cases along track, where NX represents the number of cases occurring for
scenario SX, then we compute DARE$_{cloudy}$ as follows:
$$DARE_{cloudy} = \frac{DARE_{S1}^1 + \cdots + DARE_{S1}^{N1} + DARE_{S2}^1 + \cdots + DARE_{S2}^{N2} + DARE_{S3}^1 + \cdots + DARE_{S3}^{N3}}{N1 + N2 + N3} \qquad (3)$$
When clouds are absent in the atmosphere, DARE$_S$ of aerosol in clear skies (i.e., DARE$_{non-cloudy}$ for scenario S4) is the
subtraction of upward fluxes for clear skies without aerosols and clear skies with aerosol present. If we have N4 x S4
cases, we compute DARE$_{non-cloudy}$ as follows:
$$DARE_{non-cloudy} = \frac{DARE_{S4}^1 + \cdots + DARE_{S4}^{N4}}{N4} \qquad (4)$$
To be consistent with the assumptions in the RT used for the MODIS COT retrieval (i.e., MODIS assumes CF=1 to
retrieve COT), we assign MODIS CF values of 1 for S1, S2, and S3 and 0 for S4. Finally, we compute DARE in all
sky conditions (DARE$_{all-sky}$ for scenario S1, S2, S3 and S4) as follows:
$$DARE_{all-sky} = \frac{DARE_{cloudy} \times (N1 + N2 + N3) + DARE_{non-cloudy} \times N4}{N1 + N2 + N3 + N4} \qquad (5)$$

**2.1.4 DARE$_S$ Uncertainties**
We vary AOD, CWP, SSA, ASY and surface albedo according to their uncertainties in our DARE$_S$ calculations to
obtain the DARE$_S$ uncertainties. Table 4 describes the assumed or computed uncertainties used on these five input
parameters to DARE$_S$.

| Variable | Uncertainty |
|---|---|
| CALIOP$_{ACAOD\_standard}$, CALIOP$_{AOD\_standard}$ or CALIOP$_{ACAOD\_DR}$ | Computed using Eq. (6-7) |
| CALIOP$_{ODAOD}$ | 0.11(Ryan et al., 2024) |
| MODIS$_{Cloud}$ CWP | Reported at pixel-level (Platnick et al., 2021) |
| MERRA-2 SSA | 0.05 (e.g., Jethva et al., 2024) |
| MERRA-2 ASY | 0.02 (e.g., Kassianov et al., 2012) |
| Surface Albedo | 0.01 (Jin et al., 2011) |





**Table 4: Input uncertainties on AOD, CWP, SSA, ASY and surface albedo used in our DAREs uncertainty**
**calculation. See Table 1 for definition of CALIOP$_{ACAOD\_standard}$, CALIOP$_{AOD\_standard}$, CALIOP$_{ACAOD\_DR}$,**
**MODIS$_{Cloud}$ and MERRA-2 and Table 2 on how these input parameters are used to computed DAREs.**

Regarding uncertainties on the AOD values, we use Eq. 6 and 7 described below to compute an uncertainty on
CALIOP$_{ACAOD\_standard}$, CALIOP$_{AOD\_standard}$ and CALIOP$_{ACAOD\_DR}$ for each 20km stretch. We assume a gaussian
distribution of N quantity of single-shot samples $x_1$, …, $x_N$ (e.g., CALIOP$_{ACAOD\_standard}$, CALIOP$_{AOD\_standard}$ or
CALIOP$_{ACAOD\_DR}$) with each sample $x_i$ recording a single shot uncertainty $\sigma_i$ reported by the CALIOP team. We
compute a weighted mean $\mu$ over a 20km stretch as follows:
$$\mu = \frac{\Sigma_1^N\left(\frac{x_i}{\sigma_i^2}\right)}{\Sigma_1^N\left(\frac{1}{\sigma_i^2}\right)} \qquad (6)$$

where the weighting factor is the inverse square of the error,$1/\sigma_i^2$. Note that the smaller the uncertainty, the larger the
weight and vice-versa. The error on the weighted mean can be computed as follows:
$$\sigma^2(\mu) = \frac{1}{\Sigma_1^N\left(\frac{1}{\sigma_i^2}\right)} \qquad (7)$$

(Bevington and Robinson, 1992). When filtering for ocean surface wind speeds between 3 and 15 m.s$^{-1}$ in Ryan et al.,
(2024), CALIOP$_{ODAOD}$ values have an averaged uncertainty of ~0.11 ± 0.01 (75 % ± 37 % relative) day and night.
This uncertainty is mostly due to ocean surface wind speed. In our study, we average CALIOP$_{ODAOD}$ over 20km
stretches, for which the ocean surface wind speed remains constant because we use MERRA-2 with a horizontal
resolution of ~55km. Therefore, in our study, we use a constant value of 0.11 for the averaged uncertainty on
CALIOP$_{ODAOD}$. We use reported uncertainties at the pixel-level on CWP (Platnick et al., 2021). As for uncertainties
on the aerosol intensive properties, we use an uncertainty of 0.05 for SSA and 0.02 for ASY. The averaged SSA
uncertainty of 0.05 is inspired by Jethva et al. (2024), who developed a novel synergy algorithm that combines direct
airborne measurements of above-cloud aerosol optical depth and the TOA spectral reflectance from Ozone Monitoring
Instrument (OMI) and MODIS sensors. It shows, in its Table 3, a maximum absolute uncertainty of -0.054 in the
retrieved near-UV SSA for an error of −40 % (underestimation) in ACAOD results. The averaged ASY uncertainty of
0.02 is inspired by figure 9 in Kassaniov et al. (2012) that investigates the expected accuracy of 4STAR. It describes
the relative difference between "true" and retrieved values of ASY for four selected days and shows a maximum of
+-0.02 uncertainty for ASY at 1.02 μm, which becomes smaller at shorter wavelengths. Finally, we assume an
averaged uncertainty of 0.01 in the surface albedo, inspired by figure 10 in Jin et al., (2011) in which they find a
standard deviation of ~0.01 between measured and parameterized broadband shortwave albedo for two years (2000-

464 2001).


For each one-km stretch, we compute DAREs using the uncertainty ranges of each variable (see Table 4), compute
upper and lower bounds for DAREs and then combine these values to get the DAREs uncertainty.





While designing our algorithm, we have evaluated the effects of a few constraints in the computation of our final
DARE$_S$ results (identified with an asterisk in Table A3). We separate these effects, reported in Table A4 of the
appendix, in five categories – the effects of (i) adding a lower threshold on extinction coefficients, (ii) adding aerosol
information below clouds, (iii) spatially extending aerosol top height information, (iv) using AOD version 2 along
track, (v) using AOD version 3 along the track and (vi) using corrected clouds instead of uncorrected clouds for
overlying aerosols.
Regarding category (iv) and (v), we have computed DARE$_S$ using three AOD versions (we call these V1, V2 and V3
– see third section of Table A3) and have evaluated the differences it makes in the number of 1km-data points and
DARE$_S$ results. The latitudinal evolution of AOD V1, V2, and V3 along the CALIOP track are illustrated in Fig. A4
in the appendix. Using AOD V2 instead of AOD V1, adds up to N=65 1km-data points in Table A4 and makes a
difference in mean instant all-sky (S1-S4) DARE$_S$ of maximum ~1.3 W·m$^{-2}$. Using AOD V3 instead of AOD V2,
makes the most difference in mean instant all-sky DARE$_S$ (i.e., up to 3.2 W·m$^{-2}$ difference in DARE$_S$ on 08/13/2017).
Regarding category (vi), using corrected vs. uncorrected clouds paired with AOD above them >0.3 leads to a
difference in mean instant above-thick cloud (S1) DARE$_S$ values up to ~4.1 W·m$^{-2}$ on 09/20/2017 (comparison shown
in more detail in Fig. A5 and Table A5 in the appendix). Applying a correction to the clouds does not seem to matter
much in our study regarding DARE$_S$ or COT. We argue that this is likely due to the generally low AOD values on all
three days. Note that we would probably notice a significant difference in DARE when clouds are corrected vs.
uncorrected if we were to apply our DARE calculations to multiple years over the region of Southeast Atlantic. We
emphasize that all the DARE$_S$ results in section 3 use a lower threshold on the extinction coefficients, aerosol
information below clouds, extended aerosol top heights, AOD version 2 and clouds that are not corrected for aerosol
above them.
**2.2 Parametrized DARE$_P$ Calculations**
The DARE$_P$ (see Table 2) parametrization framework in this section was developed by Cochrane et al., (2021) (see
their Eq. (12)). It collectively used airborne observations from the ORACLES field campaigns over Southeast Atlantic
in conjunction with DARE calculations to derive statistical relationship between a) DARE and b) aerosol and cloud
properties. This allows the DARE of the entire ORACLES campaign to be generalized into a minimal set of
parametrizations. Specifically, for a range of SZAs, within the Southeast Atlantic region and during ORACLES (nine
cases from the 2016 and 2017 ORACLES deployments to be exact), it links a broadband instantaneous DARE$_P$
estimate for typical biomass burning aerosols injected above an omnipresent stratocumulus deck and in clear skies to
two driving parameters, which are (i) a measure of the AOD (i.e., in our study, a combination of CALIOP$_{ACAOD\_standard}$,
CALIOP$_{ACAOD\_DR}$, CALIOP$_{AOD\_standard}$, and CALIOP$_{ODAOD}$ at 532 nm) and (ii) a measure of the albedo of the
underlying surface (i.e., either clouds or the ocean surface). Cochrane et al., (2021) report that their parametrization
leads to 20% DARE$_P$ uncertainty (lower bound on DARE variability) and this uncertainty is due to factors other than
AOD and scene albedo, such as measurement uncertainty and natural variability of the cloud and aerosol properties.
Note that, had we had a satellite retrieval of SSA with minimal uncertainty, we could have reduced the uncertainty of
DARE$_P$ by using the second parametrization in Cochrane et al., (2021) that requires SSA in addition to the AOD and



the scene albedo. The advantage of $DARE_P$ is that it establishes a direct link between DARE and two driving
parameters, and it circumvents the need for radiative transfer calculations, aerosol composition, aerosol and cloud top
height, atmospheric profiles or ocean surface wind information that are required to compute semi-observational
$DARE_S$ in our study (see Table 2). We emphasize that this parametrization only represents the relationship between
DARE and aerosol and cloud properties of the ORACLES study region as sampled and cannot be used outside of this
framework. For example, aerosol and cloud types would vary over the other regions of the globe or in different seasons
(spatial and temporal limitations), which could alter the DARE to cloud and aerosol relationship. In section 3.3.1, we
compare instantaneous $DARE_S$ and $DARE_P$ as a first way to evaluate our results.

**3 Results**
First, we describe the atmospheric scenes during three suborbital ORACLES flights (section 3.1). Second, we analyze
the temporal and spatial variability of aerosol, cloud properties, and all-sky $DARE_S$ during these suborbital flights
(see section 3.2). Third, we evaluate $DARE_S$ results (section 3.3) using two methods – collocated $DARE_P$ (section
3.3.1) and airborne SSFR upward spectral irradiance measurements (section 3.3.2).

**3.1 Suborbital Flights for Evaluation**
Figure 2 illustrates our three case studies offshore from Namibia, South Africa. The MODIS RGB images in Fig. 2
show an omnipresent stratocumulus deck on all three days but a variability in cloud types along the CALIOP track
(i.e., broken, uniform, thick and/ or thin – see Table 3). It also shows aerosol plumes of different loading on all three
days with CALIOP AOD overlaid along the track from 0.01 (dark blue) to above 1 (dark red). On each day, both a
high-flying plane focusing on remote sensing and low-flying plane focusing on in-situ sampling were deployed
(Redemann et al. 2021). By 18 and 20 September 2016, strengthened westward free-tropospheric winds dispersed
aerosol broadly over the stratocumulus deck, up to an altitude of 6.0 km (Redemann et al., 2021; Ryoo et al. 2022).
The highest aerosol loadings of ORACLES-2016 were recorded on 20 September (Pistone et al., 2019; Redemann et
al., 2021). The highest AOD values (i.e., >0.5) are clearly visible on 20 September 2016 in Fig. 2. The aerosol loadings
were a maximum during the day, and diminishing towards sunrise and sunset (Ryoo et al., 2022). The SSA is
approximately 0.85 on 20 September 2016 at a wavelength of 500 nm, based on both in-situ and SSFR retrievals
(Pistone et al. 2019). During 13 August 2017, the aerosol was located lower, within a drier (RH<60%) layer with its
top at 3 km, resting on top of a thinner cloud deck transitioning from overcast to broken. Smoke aerosol was also
sampled in the boundary layer on 13 August 2017 (Zhang and Zuidema, 2019). The CALIOP track is well aligned
with the high-altitude ER-2 aircraft (in light blue) on the two first days and with the lower-altitude P3 aircraft (in
darker blue) on the third day, which were purposely achieved by flight plannings beforehand. Among the instruments
flying onboard the ER-2, the HSRL-2, RSP and eMAS instruments are usually used to evaluate aerosol and cloud
properties retrieved from CALIOP and/ or MODIS. Note that we do not use measurements from these airborne
instruments in this paper. Among the many instruments flying on board the P3 aircraft, our focus is on the Solar
Spectral Flux Radiometer (SSFR) instrument (Pilewskie et al., 2003; Schmidt and Pilewski, 2012) which we use to
evaluate our $DARE_S$ results in section 3.3.2. We remind the reader that in this paper, we use the observations, and the




modeled parameters listed in Table 2 (i.e., MODIS for cloud microphysics, CALIOP for AOD, cloud and aerosol
heights, MERRA-2 for aerosol intensive properties, atmospheric profiles and winds) to compute all-sky SW TOA
DAREs for each 1km stretch along the CALIOP track on each day.


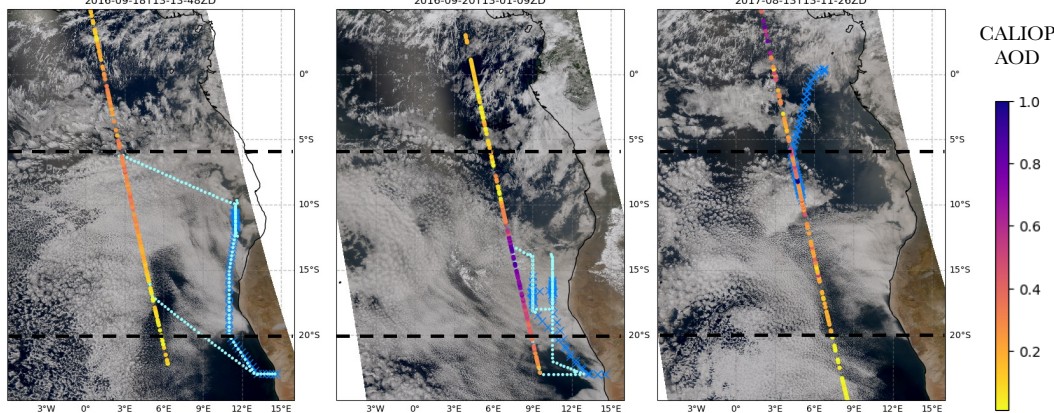



**Figure 2: Three case studies during ORACLES offshore from Namibia, South Africa on 09/18/2016, 09/20/2016, and**
**08/13/2017. The color bar shows AOD across the CALIOP/ CALIPSO flight tracks. The AOD is described under version 2**
**in Table A3 of the appendix. MODIS RGB Rayleigh scattering-corrected reflectance is in the background, together with**
**ER-2 and P3 flight tracks in light and dark blue. 09/18/2016 and 09/20/2016 show satisfying colocation with the ER-2**
**aircraft. 08/13/2017 show satisfying colocation with the P3 aircraft.**

Figure 3 illustrates the number of S1, S2, S3 and S4 scenarios along the tracks on all three days of Fig. 2 when focusing
between 6ºS and 20ºS in latitude (see dashed horizontal black line on Fig. 2). We note a dominance of aerosol above
thick clouds (S1), followed by unassigned (N/A) cases and lastly, clear skies (S4), aerosol above and below thick, thin
and/ or broken clouds (S2 and S3).












**Figure 3: Number of S1-S4 samples on 09/18/2016, 09/20/2016 and 08/13/2017 (red, green, and blue) during ORACLES between 6ºS and 20ºS in latitude (see dashed horizontal black line on Fig. 2). S1: Thick and uniform cloud with MODIS COT>4; S2: Thick, can be broken cloud with MODIS COT>4; S3: Thin, can be broken cloud with MODIS COT<4; S4: Clear skies can contain small broken clouds (MODIS cloud mask = "clear"). See Table 3 for more details on S1-S4. N/A denotes the number of cases that were not assigned a scenario S1-S4 for various reasons (see reasons for these cases in the text)**

Any scenario labelled "N/A" on Fig. 3 is a scenario that is not assigned to any of the S1 through S4 cases. These "N/A" scenarios constitute 26, 43, and 48% of the entire number of 1km profiles on 09/18/2016, 09/20/2016 and 08/13/2017 (i.e., N=400, 673 and 754 compared to N=1560 from 6ºS to 20ºS), These scenarios could be unassigned in our study due to (i) more than one cloud present above a 3km altitude (e.g., cirrus clouds present over LWLC), (ii) less than a high confidence in the phase of LWLC (e.g., CALIOP fails to classify the cloud as LWLC or CALIOP successfully classifies the cloud as non-LWLC), (iii) a disconnect between CALIOP and MODIS-based cloud characterization and/ or clear sky determination (e.g., CALIOP points at a "thick cloud" along the CALIOP track and MODIS points at a cloud with COT<4 within the 1km pixel or CALIOP points at mostly clear skies along the CALIOP track and MODIS points at mostly cloudy skies within the 1km pixel), and/ or (iv) CALIOP-based cloudy skies present but no collocated valid MODIS COT retrieval.

### 3.2 Aerosol, Cloud Properties, and All-Sky DARES

Figure 4 shows the Probability Distribution Functions (PDFs) of the AOD above clouds (S1-S3), AOD in clear skies (S4), all-sky SSA, ASY and EAE (S1-S4) values of the aerosol layer at the highest altitude, COT, CWP and CER of all clouds (S1-S3), and diurnal mean DARES values for all three days (red, green and blue). Table 5 shows the mean



values corresponding to the PDFs of Fig. 4 as well as other parameters such as DARE, uncertainties on DARE (instant
and 24h), uncertainties on CWP and AOD, CF, ATH and ABH. Table A6 in the appendix complements Table 5 by
providing the same parameters for scenarios S1, S2 and S3 separately.

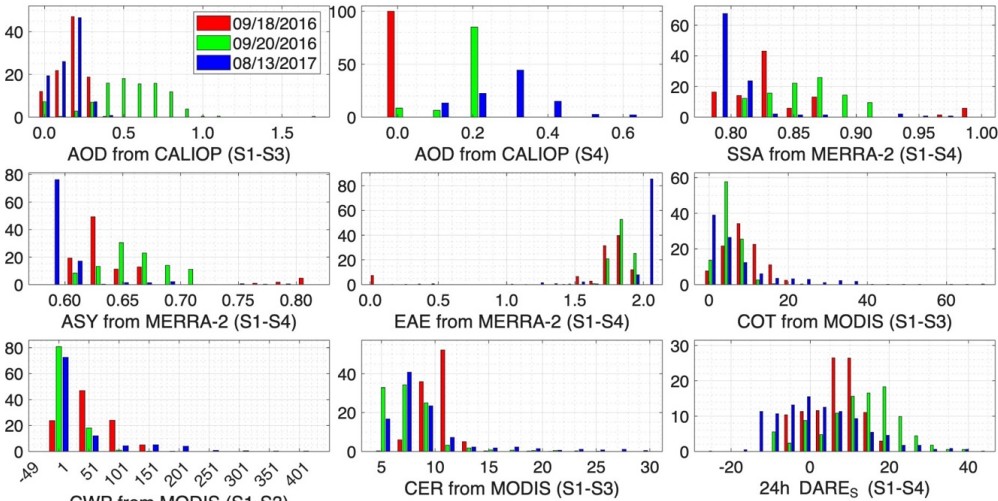

**Figure 4: Probability distribution function of Aerosol, Cloud and diurnal mean DARE$_S$ properties on 09/18/2016,**
**09/20/2016 and 08/13/2017 (red, green, and blue) during ORACLES. Y-axis is the number of points in each bin. Table 5**
**shows the averaged values on each day in clear-sky, cloudy and all-sky conditions. Cloud retrieved optical properties are**
**not corrected for aerosols above them. Latitudes are selected between 6ºS and 20ºS. AOD is at 532nm; SSA and ASY are in**
**the 442-625nm RRTMG channel; EAE is computed between the 442-625nm and the 625-778nm RRTMG channels; SSA,**
**EAE and ASY are at the highest aerosol height in MERRA-2.**




| Averaged Values | 9/18/16 | | | 9/20/16 | | | 8/13/17 | | |
|---|---|---|---|---|---|---|---|---|---|
| | Clear-sky (S4) | Cloudy Skies (S1-S3) | All-sky (S1-S4) | Clear-sky (S4) | Cloudy Skies (S1-S3) | All-sky (S1-S4) | Clear-sky (S4) | Cloudy Skies (S1-S3) | All-sky (S1-S4) |
| Number | 93 | 1067 | 1160 | 47 | 840 | 887 | 187 | 619 | 806 |
| DARE 24h | -2.4 | 9.4 | 8.4 | -6.4 | 15.5 | 14.3 | -8.9 | 7.9 | 4.0 |
| ΔDARE 24h | 0.3 | 0.1 | 0.1 | 0.4 | 0.1 | 0.1 | 0.2 | 0.2 | 0.1 |
| DARE Instant | -5.1 | 22.5 | 20.3 | -15.6 | 36.9 | 34.2 | -22.6 | 19.2 | 9.5 |
| ΔDARE Instant | 0.7 | 0.2 | 0.2 | 1.0 | 0.3 | 0.3 | 0.6 | 0.4 | 0.3 |
| COT | | 10.8 | | | 7.0 | | | 9.1 | |
| CWP | | 80.3 | | | 37.5 | | | 54.2 | |
| ΔCWP | | 0.5 | | | 0.5 | | | 1.1 | |
| CER | | 11.3 | | | 8.6 | | | 10.0 | |
| CALIOP CF | 0.05 | 0.99 | | 0.04 | 0.99 | | 0.01 | 0.97 | |
| MODIS CF | 0.04 | 0.99 | | 0.05 | 0.99 | | 0 | 0.93 | |
| AOD above clouds | | 0.2 | | | 0.6 | | | 0.2 | |
| AOD in clear-sky | 0.1 | | | 0.2 | | | 0.3 | | |
| ΔAOD | 0.0 | 0.0 | | 0.0 | 0.0 | | 0.0 | 0.0 | |
| SSA at highest altitude | 1.0 | 0.8 | | 0.9 | 0.9 | | 0.8 | 0.8 | |
| ASY at highest altitude | 0.8 | 0.6 | | 0.7 | 0.7 | | 0.6 | 0.6 | |
| EAE at highest altitude | 0.2 | 1.9 | | 1.9 | 1.9 | | 2.0 | 2.0 | |
| Aerosol Top Height | 1.1 | 4.6 | | 3.5 | 5.1 | | 2.9 | 2.9 | |
| Cloud Top Height | | 1.0 | | | 0.7 | | | 1.0 | |

**Table 5: Averaged DARE values and corresponding averaged aerosol and cloud input values in the case of clear-sky (S4),**
**cloudy (S1-S3) and all-sky (S1-S4) scenarios. Some of the all-sky averaged values correspond to the PDFs in Fig. 6. We**
**display results corresponding to AOD version 2. ΔSSA is fixed at 0.05 and ΔASY is fixed at 0.02 (see Table 4). Latitudes are**
**selected between 6ºS and 20ºS**

Fig. 4 shows a variability of diurnal mean all-sky (S1-S4) DAREs from -25 to 40 W·m$^{-2}$ on all three days. It also
shows the lowest all-sky 24h DARE values are on 08/13/2017 (in blue). Table 5 also shows mean 24h DAREs of ~4
± 0.1 W.m$^{-2}$. This finding can be explained by 08/13/2017 also showing the highest number of clear-sky (S4) cases in





Fig. 3, the lowest mean CALIOP CF values in clear-sky and the highest mean AOD value in clear-sky (0.3) in Table

611     5.

Mean 24h cloudy (S1-S3) $DARE_S$ is ~9 ± 0.1, 15 ± 0.1 and 8 ± 0.2 W·m$^{-2}$ on 09/18/2016, 09/20/2016 and 08/13/2017
in Table 5. These values are higher than in Kacenelenbogen et al., (2019), where we found mean 24h cloudy DARE
values of 2.49 ± 2.54 and 2.87 ± 2.33 W·m$^{-2}$ respectively in JJA and SON over a region between 19º and 2ºN and
10ºW and 8ºE, using satellite data from 2008 to 2012. We attribute this difference in cloudy $DARE_S$ to a difference in
the period, the spatial domain, and the way $DARE_S$ is computed. Here, the highest mean 24h cloudy (S1-S3) $DARE_S$
value of ~15 W.m-2 is explained by the highest mean AOD above clouds of 0.6. But in general, the AOD above clouds
on all 3 days is like the monthly average of 0.2-0.6 in Sept 2016 and Aug 2017 in Chang et al., (2023) (see their Fig.
1). Also, Doherty et al., (2022) (see their Table 3) shows a monthly average of integrated vertical profiles of scattering
and absorption coefficients above clouds from in-situ instruments of 0.4 in 2016 and 0.3-0.6 in 2017. Table 1 in
Kacenelenbogen et al. (2019) lists other peer-reviewed calculations of cloudy DARE to which our results can be
compared (e.g., Chand et al., 2009, Wilcox 2012, De Graaf et al., 2012, 2014, Meyer et al., 2013, 2015, Peers et al.,
2015, and Feng and Christopher 2015)).

Figure 5 illustrates the spatial evolution of key input parameters to our (instantaneous and diurnal mean) $DARE_S$
calculations, together with the $DARE_S$ values themselves along the CALIOP track on 08/13/2017. Figure A6 and A7
in the appendix show similar plots for the two other days. From the top to the bottom panels, it shows the location of
S1, S2, S3 and S4 cases, the AOD (above cloud and in clear skies) ± ΔAOD, COT, CER and CWP ± ΔCWP values
along the CALIOP track and $DARE_S$ ± Δ$DARE_S$ (24h and instantaneous). The low diurnal mean $DARE_S$ value of ~
4W·m$^{-2}$ on 08/13/2017 (see Table 5) is in fact accompanied by strong $DARE_S$ variability along the track as illustrated
on Fig. 5. For example, a thick cloud is detected at ~10ºS latitude (see also Fig. 2), which corresponds to a peak in
COT and CWP (but not CER) values. Over this region, our algorithm detects many S1 cases (in red) for which the
AOD and SSA both remain ~constant (i.e., a light absorbing aerosol plume with a strong loading) and the COT values
increase. This leads to a sharp increase in the $DARE_S$ values (i.e., more warming of the atmosphere).



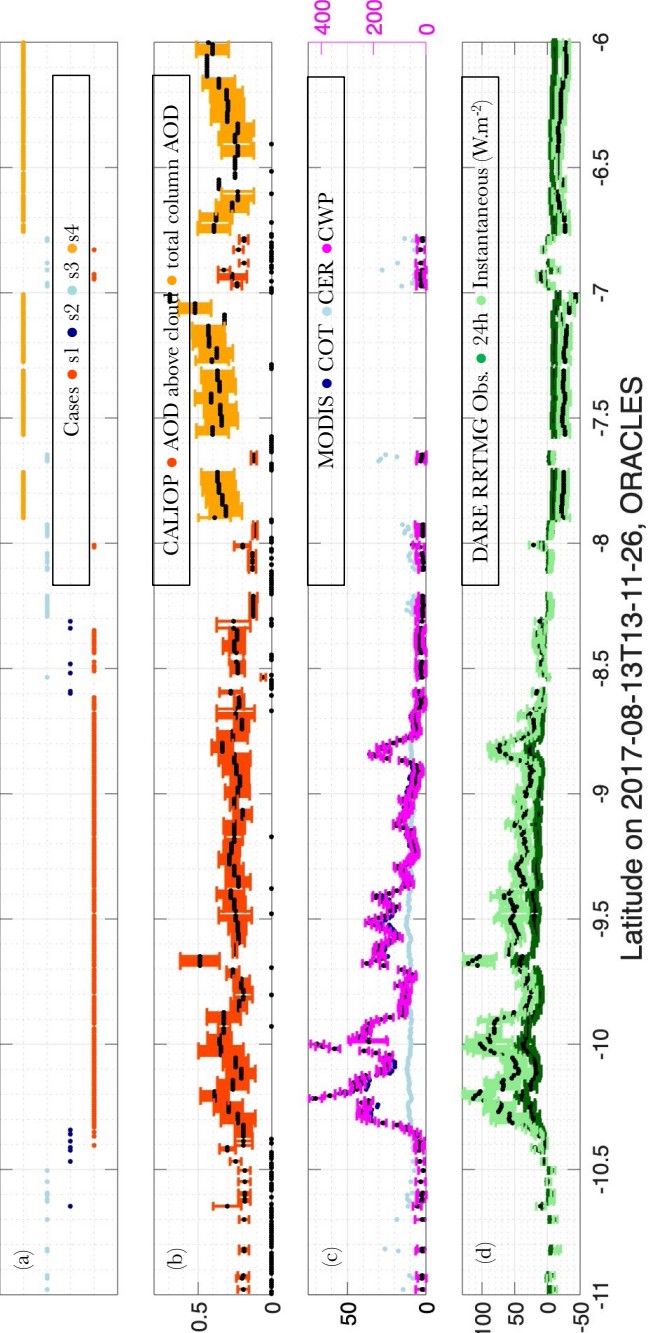

**Figure 5: Spatial evolution of key input parameters to our DARE$_S$ calculations, together with the DARE$_S$ values themselves (diurnal and instantaneous) along the CALIOP track on 08/13/2017. From the top to the bottom panel – (a) S1, S2, S3 and S4 cases, (b) the V2 AOD ±ΔAOD, (c) COT, CER, CWP ±ΔCWP, (d) 24h and instant DARE$_S$ ±ΔDARE$_S$ (W·m⁻²). Cloud retrieved optical properties are not corrected for aerosols above them. Instead of showing latitudes between 6°S and 20°S, we reduce the latitude range here from 6°S to 11°S for visibility. See Fig. A6 and A7 in the appendix for similar figures on the two other days.**





Figure A8 in the appendix illustrates 24h DARE$_S$ values as a function of AOD and SSA in clear sky conditions (i.e., S4) on the right and as a function of AOD and COT in cloudy conditions (i.e., S1-S3) on the left on 09/18/2016, 09/20/2016 and 08/13/2017. First, as expected, DARE$_S$ values are more and more negative when paired with increasing AOD values in clear skies and any SSA values. Second, also as expected, we observe a clear increase in positive DARE$_S$ values when paired with an increase of AOD values above clouds. In cloudy conditions and when the AOD above clouds remains similar, DARE$_S$ records consistently higher values (more warming) when paired with a larger COT value. Note that we were able to reproduce this relationship in our theoretical calculations (see Fig. A2).

**3.3 Assessment of DARE$_S$**

**3.3.1. Using DARE$_P$**

We first assess DARE$_S$ using the DARE$_P$ calculations described in section 2.2 and Table 2. The DARE$_P$ parametrization was developed during ORACLES, an airborne field campaign specifically designed to investigate aerosols above clouds. Because the DARE$_P$ parameterization applies only to the subset of cloudy scenarios (i.e., S1-S3) measured during ORACLES, we do not include DARE$_S$ vs. DARE$_P$ comparisons in clear sky conditions (i.e., S4) in this paper. Figure 6 shows instant DARE$_P$ on the x-axis and instant DARE$_S$ on the y-axis, coloured by the AOD values above clouds on 09/18/2016 (left), 09/20/2016 (middle) and 08/13/2017 (right). The black crosses denote CALIOP cloud fractions that are below 1. The first section of Table 6 summarises the statistics.

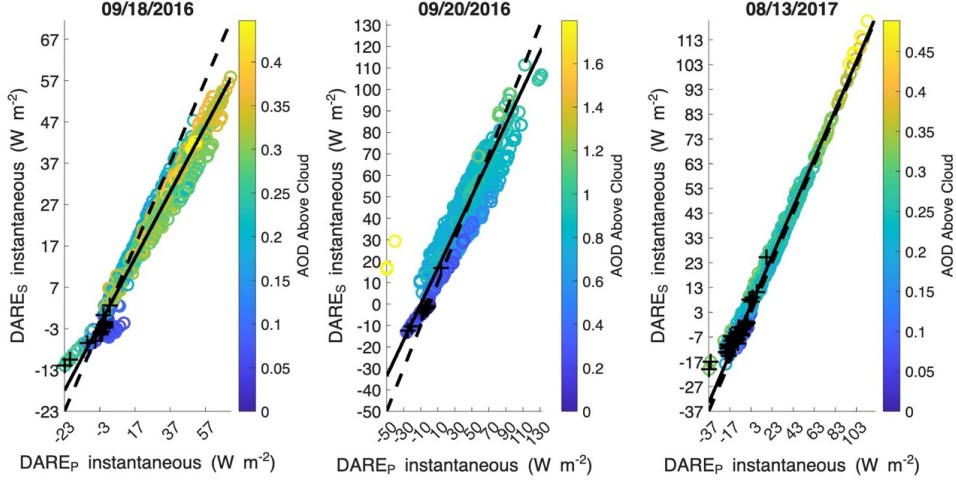

**Figure 6: Semi-observational instantaneous DARE$_S$ (y-axis) compared to DARE$_P$ (x-axis) (see section 2.2 and Table 2) for cloudy cases (i.e., S1, S2 and S3 in Table 3) at 532 nm (i.e., in the 442-625 nm channel). Cloud retrieved optical properties**



**are not corrected for aerosols above them. Latitudes are selected between 6ºS and 20ºS. Black crosses show points with**
**CALIOP cloud fraction below 1 (reflective of more broken cloud). See Table 6 for statistics.**
When evaluating our semi-observational $DARE_S$ with coincident parametrized $DARE_P$ over the region, we find a
satisfying agreement for cloudy DARE ($R^2$=0.87 to 0.99, slope=0.80 to 0.99, offset =0.37 to 8.30, N=619 to 1067 in
Table 6). On 09/18/2016, we note that $DARE_P$ tends to slightly overestimate $DARE_S$ for high AOD above clouds.
This might be due to $DARE_P$ assuming only one aerosol layer, and/ or erroneously simulated MERRA-2 SSA and
ASY values in our $DARE_S$ calculations. For example, when computing $DARE_T$ (see Fig. A2), we record lower DARE
values (by ~10 W m$^{-2}$) when adding more scattering aerosols (i.e., "continental") to already absorbing aerosols (i.e.,
"urban") over a thick cloud (COT=10). We also note a distinctive feature on Fig. 6 on 09/18/2016 away from the 1:1
line for low AOD and CALIOP cloud fractions below 1 (black crosses). This feature is very likely due to cloud
homogeneities paired with low AOD values.



| (1) Cloudy (S1-S3) DARE$_P$ vs. DARE$_S$ at 532nm | Dates | | |
|---|---|---|---|
| | 09/18/2016 | 09/20/2016 | 08/13/2017 |
| Number | 1067 | 840 | 619 |
| RMSE | 6.9 (31%) | 10.3 (28%) | 3.6 (19%) |
| $R^2$ | 0.97 | 0.87 | 0.99 |
| Slope | 0.8 | 0.8 | 1.0 |
| Offset | 0.4 | 8.3 | 2.9 |
| Mean cloudy DARE$_P$ | 27.7 | 34.1 | 16.6 |
| Mean cloudy DARE$_S$ | 22.5 | 36.9 | 19.2 |

| (2) All-sky (S1-S4) SSFR vs. DARE$_S$-related fluxes on 08/13/2017 | SW RRTMG Bands (nm) | | | |
|---|---|---|---|---|
| | 778-1242 | 625-778 | 442-625 | 345-442 |
| Number | 51 | | | |
| RMSE | 17.4 (17%) | 6.9 (9%) | 11.7 (9%) | 8.2 (16%) |
| $R^2$ | 0.95 | 0.95 | 0.95 | 0.94 |
| Slope | 1.1 | 0.9 | 0.8 | 0.8 |
| Offset | 7.0 | 5.8 | 14.9 | 13.9 |
| Mean SSFR-measured fluxes | 89.7 | 78.7 | 128.5 | 43.3 |
| Mean DARE$_S$-related fluxes | 104.9 | 77.0 | 124.4 | 50.8 |

**Table 6: Number, Root Mean Square error (RMSE), correlation coefficient, $R^2$, and linear regression parameters between**
**(1) cloudy (S1-S3) DARE$_P$ vs. DARE$_S$ at 532nm (i.e., in the 442-625nm channel) for our three case studies and (2) all-sky**
**(S1-S4) SSFR-measured and DARE$_S$-related fluxes in four SW RRTMG broadband channels; % in parathesis is based on**
**the (1) mean cloudy DARE$_S$ and (2) mean DARE$_S$-related related fluxes. Latitudes are between 6ºS and 20ºS.**





### 3.3.2. Using Airborne SSFR Upward Spectral Irradiance Measurements

After the statistical assessment of DARE$_S$ using DARE$_P$ in section 3.3.1, we now assess DARE$_S$ using the spatially and temporally co-located SSFR measurements on our third case study of 08/13/2017 (see Fig. 2). Although the collocation only provides limited samples for validation, the directly measured irradiance (which can be used to indicate radiative effects) from SSFR can provide further insights in our DARE$_S$ results.

We consider only the locations and times when (i) the aircraft flies above the CALIOP-inferred aerosol top height, (ii) the aircraft measurements are within ($\leq$) 0.7 km, (iii) the aircraft measurements are within $\pm$ 30 min of the CALIOP observations (i.e., between 13:00 and 14:00 UTC as the overpass occurs at ~13:30 UTC over the region) and (iv) the aircraft is leveled (i.e., the aircraft pitch and role are both within $\pm$ 5 degrees). After applying those filters, we find N=51 valid (>0) paired CALIOP-SSFR flux results corresponding to aircraft altitudes ranging between 3.57 and 6.46 km above CALIOP aerosol top heights ranging from 3.05 to 3.14 km, distances between CALIOP and the nearest SSFR measurements ranging from 0.44 to 0.70 km, times of SSFR measurements between 13:14 and 13:55 UTC, joint latitudes between 7.86ºS and 9.56ºS (see Fig. 2 for context) and aircraft pitch and role between -1.5 and 3.5º. We use SSFR files called "20170813_calibspecs_20171106p_1324_20170814s_150C_attcorr_ratio.nc" and "20170813_librad_info.nc".

As a reminder, DARE$_S$ is the subtraction of the upward spectral broadband irradiances (or fluxes received by a surface per unit area), $F_{b\lambda}\uparrow_{\text{no aerosol}}$ - $F_{b\lambda}\uparrow_{\text{aerosols}}$ in 13 RRTMG broadband channels (from 200 nm to 3,846 nm with spectral bands ranging from 56 to 769 nm in W·m$^{-2}$). The airborne SSFR instrument measures the upward flux, $F_\lambda\uparrow$, in narrow spectral bands (from 350 nm to 2,200 nm with spectral resolution of 6 nm to 12 nm in W·m$^{-2}$·nm$^{-1}$). We spectrally integrate SSFR $F_\lambda\uparrow$ within each SW RRTMG broadband channel using a trapezoidal numerical integration. For example, the first RRTMG channel that contains SSFR measurements is between 345nm and 442nm. SSFR's shortest channel is at 350nm and measures 15 increments of 6nm-spaced $F_\lambda\uparrow$ up to 442nm i.e., within the first RRTMG channel. Therefore, we sum all 15 increments of $F_\lambda\uparrow$ from SSFR (i.e., from 350nm to 442nm) and compare this value to $F_{b\lambda}\uparrow$ in the first RRTMG channel (i.e., from 345 to 442nm). The second part of Table 6 shows a satisfying agreement between SSFR $F_\lambda\uparrow$ and $F_\lambda\uparrow$ at the source of our semi-observational DARE$_S$ in four relevant RRTMG broad band channels (i.e., 345-442, 442-625, 625-778 and 778-1242nm). This is illustrated by a high correlation coefficient (0.94-0.95) and an RMSE value between 9 and 17 % of the mean $F_\lambda\uparrow$ at the source of our semi-observational DARE$_S$ in Table 6. Fig. A9 in the appendix shows the comparison between SSFR-measured and DARE$_S$-related fluxes as a function of distance between the aircraft and the satellite track. Figure 7 is like Fig. 5 but focuses on the comparison between collocated airborne and satellite observations (i.e., from -9.6 to -7.9º Latitude). Panel (a) shows the N=51 collocated cases with valid satellite and airborne data (see black crosses) and the different S1-S4 scenarios as a function of latitude. Among our N=51 points, we find a majority of S1 cases, followed by S3 and S4 cases in this stretch. Panel (b) shows AOD $\pm\Delta$AOD above clouds and in clear skies. Panel (c) shows COT, CER and CWP $\pm\Delta$CWP. Panel (d) shows the satellite radiative fluxes, $F_{b\lambda}\uparrow$, behind our DARE$_S$ calculations in light green (W·m$^{-2}$) and the distance between the aircraft and the CALIOP ground track in magenta from 0.45 to 0.70 km. Panel (e) shows the



absolute difference between SSFR and satellite $F_{b\lambda}\uparrow$ as a percentage of the satellite radiative fluxes in all four
broadband channels (778-1242 in solid grey, 625-778 in solid black, 442-625 in dotted black, 345-442 nm in dotted
grey).
From ~9.2ºS to 7.9ºS in latitude in Fig. 7, distances between the aircraft and the CALIOP ground track are higher
(>600m in magenta in (d)), clouds are thinner (i.e., low COT values in dark blue in (c)) and/ or more broken (i.e.,
more S3 cases in (a)), and AOD above cloud is smaller (in red in (b)). These conditions all seem to lead to more
unstable and generally higher satellite-SSFR flux differences (e). This is confirmed on Fig. A9 where we observe
more scatter between SSFR $F_\lambda\uparrow$ and $F_\lambda\uparrow$ at the source of our semi-observational $DARE_S$ when the distance between
satellite and aircraft increases (see yellow markers) in all four channels.
If we focus on points of close satellite-aircraft collocation (i.e., from ~9.6ºS to 9.2ºS, <600m in magenta in (d)),
satellite $F_{b\lambda}\uparrow$ (in light green in (d)) shows high values (>100 W m$^{-2}$) due to the presence of S1 cases (red dots in (a)),
high AOD in (b) and high COT values in (c). For these points, we find an absolute difference in all four broadband
channels below ~20% and an absolute difference below 15% between 778 and 442 nm (solid black and dotted black
in (e)).









**Figure 7. Spatial evolution of key input parameters along the CALIOP track, with a focus on when and where we have**
**collocated airborne SFFR measurements for validation on 08/13/2017 – (a) S1, S2, S3, S4 cases (red, dark blue, light blue,**
**and orange) and collocated SSFR measurements (black crosses), (b) V2 AOD ±ΔAOD (red above cloud and orange in clear-**
**sky), (c) COT (dark blue), CER (light blue), CWP±ΔCWP (magenta), (d) collocated satellite broadband (spectrally**
**integrated) upward irradiance (or flux) received by a surface per unit area in W·m⁻², $F_{b\lambda}\uparrow$, behind our DARE$_S$ calculations**
**in light green (W·m⁻²) and distance between the aircraft and the CALIOP ground track in magenta, (e) absolute difference**
**between SSFR and satellite $F_{b\lambda}\uparrow$ in all four broadband channels (solid grey for 778-1242, solid black for 625-778, dotted**
**black for 442-625, dotted grey for 345-442 nm) as a percentage of satellite $F_{b\lambda}\uparrow$. Cloud retrieved optical properties are not**
**corrected for aerosols above them.**

**4 Discussions and Future Work**
As described in Table 2, MERRA-2 is used in this paper to define the uppermost aerosol top height and lowermost
aerosol base height below clouds, the vertical distribution of spectral aerosol extinction coefficient, ASY and SSA,
and the atmospheric composition, weather and ocean surface winds. First, we currently use MERRA-2's vertical
distribution of aerosols at face value with no consideration of a very likely bias in the modelled aerosol vertical profile
(see section 2.1.2). An improvement worth exploring would be to select the MERRA-2 vertical location that
corresponds to the strongest aerosol signal. Second, another improvement would be to infer aerosol vertical
distribution and loading below clouds as a function of near-by satellite-observed clear-sky aerosol cases. Third, pairing
ESA/JAXA EarthCARE (Wehr et al., 2023) launched in May 2024, NASA PACE (Werdell et al., 2019) Spexone
(Hasekamp et al., 2019) and HARP2 (Gao et al., 2023) launched in Feb 2024 might provide some insight on the
observed vertical distribution of spectral aerosol extinction coefficient, ASY and SSA. The EarthCARE processing



chain includes operational synergistic lidar, radar, and imager cloud fields, profiles of aerosols, atmospheric heating
rates and top-of-atmosphere SW and longwave fluxes using 3D radiative transfer. These fluxes are automatically
compared with EarthCARE broad-band radiometer measurements, allowing for a radiative closure assessment of the
retrieved cloud and aerosol properties. However, let us emphasize that the PACE and EarthCARE satellites are never
perfectly co-located in both time and space. The Atmosphere Observing System mission (AOS), on the other hand,
holds promising new science as it still consists, at the time of writing, of a suite of lidar, radar, and radiometer satellites
flying in formation to jointly observe aerosol, cloud, convection, and precipitation. We note that using EarthCARE's
joint lidar and imager (possibly paired with PACE polarimeters) will likely reduce the number of unassigned scenarios
in this paper as it will provide improved LWLC classification and optical properties, and possibly reduce the mismatch
between cloudy and clear-sky scenes.
In this paper, to compute our 24h $DARE_S$, we solely vary SZAs every hour during the day, which implicitly assumes
constant aerosol and cloud vertical optical properties (see Table 2). On a global scale, most of the diurnal DARE
variability is due to the varying solar zenith angles. Global diurnal mean DARE does not need many hourly
measurements if the AOD is representative of the daily mean (e.g., at the Aqua and Terra overpass times). For example,
Arola et al. (2013) found that the average impact of diurnal AOD variability on 24h mean DARE estimates is small
when averaged over all global Aerosol Robotic Network (AERONET) sites (Holben et al., 1998). Regional DARE,
unlike global DARE, can show considerable variability throughout the day due to varying aerosol and cloud fields.
Xu et al. (2016), for example, show that the daily mean clear skies TOA DARE is overestimated by up to 3.9 W·m$^{-2}$
in the summertime in Beijing if they use a constant Aqua MODIS AOD value, compared to accounting for the
observed hourly averaged daily variability. According to Min and Zhang (2014) (see their Table 2), assuming a
constant CF derived from Aqua MODIS generally leads to an underestimation (less positive) by 16% in the all-skies
DARE calculations. Chang et al. (2025) find that including observed cloud diurnal cycle from geostationary satellites
over the southeast Atlantic results in nearly a twofold (about 1.4 W m-2) increase in the regional mean aerosol radiative
warming, compared to assuming a constant early-afternoon cloud field throughout the entire day. We plan on adding
diurnal aerosol and cloud information in our $DARE_S$ calculations (instead of only varying SZA) using co-located
geostationary satellite observations.
Another extension to this work is to add an atmospheric scenario for which we observe one or more clouds overlying
the LWLCs. With the addition of this multi-cloud atmospheric scenario, $DARE_S$ will be one step closer to a truly all-
sky TOA SW $DARE_S$. We envision this additional scenario to use (i) the CALIPSO-CloudSat-CERES-MODIS
(CCCM or C3M) (Kato et al., 2010, 2011) derived cloud heights and cloud microphysical properties and (ii) MERRA-
2 simulated aerosol extensive and intensive properties. The new all-sky $DARE_S$ results can then be evaluated using
collocated airborne field campaign observations such as from the HSRL-2 and the SFFR instruments during the Cloud,
Aerosol and Monsoon Processes Philippines Experiment (CAMP$^2$Ex) in 2019 over Southeast Asia. Note that adding
atmospheric scenes showing multiple clouds on the vertical would increase the overall number of assigned
atmospheric scenarios in our study.






At present, the order of importance of key aerosol, cloud and surface parameters in DARE calculations remains
unclear. Thorsen et al., (2020) find that in clear skies, AOD, SSA and ASY is the order of importance of key aerosol
parameters in DARE calculations. However, priorities can differ regionally according to airborne DARE sensitivity
studies (e.g., Cochrane et al., 2019, 2021). According to Elsey et al., (2024), the AOD uncertainty is the main
contributor to the overall uncertainty on DARE except over bright surfaces where SSA uncertainty contributes most.
We plan to apply our DARE$_S$ calculations to multiple years of combined satellite and model data over different regions
of the world. Three example regions over the Atlantic Ocean are the Southeast Atlantic (this paper), the North Atlantic
offshore from the Sahel, and a third region encompassing the latter two; the first two regions are dominated by different
aerosol and cloud regimes and the third one represents the transition between these two regimes. We then plan to use
this larger DARE$_S$ dataset for different atmospheric scenarios, over specific regions of the world and linked to key
cloud, aerosol and surface input parameters to assess the order of importance of these parameters in DARE$_S$
calculations for specific aerosol and cloud regimes.

**5 Conclusion**
We compute TOA SW all-sky DARE combining CALIOP, MODIS and MERRA-2 along the CALIOP track. These
computations are made for four different atmospheric scenarios of aerosols above and below thick, thin and/ or broken
clouds or aerosols in (mostly) clear skies. The clouds in our study must be single layer and low level (<3km) liquid
clouds. We focus our analysis on three days over the Southeast Atlantic for which we compare our semi-observational
DARE results to co-located suborbital aerosol and cloud observations during the ORACLES field campaign. During
these three days, satellite observations show a high number of cases with aerosols above and below thick and
homogeneous clouds (i.e., N=334-968 or 21-62% of our dataset), followed by cases that are not assigned in our study
(i.e., N=400-754 or 26-48% of our dataset).
The semi-observational diurnal average DARE values for our three days range from -25 (cooling) to 40 W·m$^{-2}$
(warming). Highly positive DARE values are mostly due to aerosols with high AOD above clouds with high COT
values. Highly negative DARE values, on the other hand, are mostly due to aerosols with high AOD values in clear-
sky cases. We use two ways of evaluating our semi-observational DARE -- a DARE parametrization, dependent on
the AOD and cloud albedo, that was designed using SSFR measurements during the ORACLES field campaign and
an upward irradiance (or flux) directly measured by the airborne SSFR instrument. First, we demonstrate agreement
between our semi-observational satellite DARE and coincident parametrized DARE over the region (R$^2$=0.97-0.99,
RMSE=19-31%, N=619-1067). Second, we also demonstrate agreement between our semi-observational satellite
upward spectral irradiance with coincident measurements from the co-located SSFR instrument in four short-wave
broadband channels during ORACLES (R$^2$=0.94-0.95, RMSE=9-17%, N=51).
We emphasize that using the EarthCARE lidar and imager instruments instead of pairing A-Train's CALIOP and
MODIS as well as adding cases with one or more clouds above our single water cloud would bring our results closer
to a truly all-sky DARE results (and drastically decrease the number of unassigned atmospheric scenarios in our study).
We also plan on adding aerosol and cloud diurnal cycle information from co-located geostationary satellites to improve



our diurnal mean all-sky semi-observational DARE$_S$ results. Finally, in this paper, we have concentrated on three case
studies to examine our methodology in detail and evaluate the results against airborne SSFR measurements. This is a
necessary first step before applying our algorithm to multiple years of combined satellite and model data over different
regions of the world. Our goal is to ultimately assess the order of importance of atmospheric parameters in the
calculation of DARE for specific aerosol and cloud regimes. Expanding on the work done in this study will inform
future missions on where, when, and how accurately the retrievals should be performed to most effectively reduce all-
skies DARE uncertainties.

**Code and data availability**
The CALIPSO Lidar Level 2 1 km Cloud Layer, V4-51 is publicly archived here:
https://asdc.larc.nasa.gov/project/CALIPSO/CAL_LID_L2_01kmCLay-Standard-V4-51_V4-51. The CALIPSO
Lidar Level 2 5 km Merged Layer, V4-51 is publicly archived here:
https://asdc.larc.nasa.gov/project/CALIPSO/CAL_LID_L2_05kmMLay-Standard-V4-51_V4-51. The CALIPSO
Lidar Level 2 Aerosol Profile, V4-51 is publicly archived here:
https://asdc.larc.nasa.gov/project/CALIPSO/CAL_LID_L2_05kmAPro-Standard-V4-51_V4-51. The MODIS
CLDPROP_L2_MODIS_Aqua - MODIS/Aqua Cloud Properties L2 5-Min Swath 1000 m is publicly archived here:
https://ladsweb.modaps.eosdis.nasa.gov/missions-and-measurements/products/CLDPROP_L2_MODIS_Aqua/.
MERRA-2 data are available at MDISC: https://disc.gsfc.nasa.gov/datasets?project=MERRA-2, managed by the
NASA Goddard Earth Sciences (GES) Data and Information Services Center (DISC).

**Author contributions**
MK designed and performed the scientific analysis and led the preparation of the manuscript with contributions from
all authors. KM, EN, MV, HC, SS, RF, RL, HB, PZ, RH and WM provided extensive guidance and insight into the
scientific analysis. EN, KM, MV, and HC provided specific data and guidance on how to use the data from MERRA-
2, MODIS, CALIOP, and SSFR respectively. RK was instrumental in designing the atmospheric scenarios and
processed the matching of CALIOP, MODIS and MERRA-2 and other supporting analysis. NA was instrumental in
processing all the RT-related calculations. MK analysed all the output results.

**Competing interests**
The contact author has declared that none of the authors has any competing interests

**Acknowledgements**
The authors are extremely grateful for guidance and technical support from the CALIOP, MODIS, MERRA-2 and
SSFR teams.



**Financial support**
This research has been supported by the National Aeronautics and Space Administration Research Opportunities in
Space and Earth Science (ROSES) CloudSat and CALIPSO Science Team Recompete program (A.26),
NNH21ZDA001N-CCST.

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




**Appendix**

**Data and Method**


| Computation | RRTMG-SW |
|---|---|
| **Cloud Detection and Characterization** | [COT =1, CER=12, CWP=8] or [COT=10, CER=12, CWP=80] |
| **Cloud Albedo** | N/A |
| **Cloud Top Height (CTH) and Cloud Base Height (CBH)** | CTH is 1km and CBH is 0.5km |
| **Uppermost Aerosol Top Height (ATH) and lowermost Aerosol Base Height (ABH)** | ATH is 5km and ABH is 1km above CTH |
| **Vertical distribution of spectral ASY** | ASY = 0.6 |
| **Vertical distribution of spectral SSA** | Spectral SSA of two built-in RRTMG aerosol types[1] is weighted by $AOD_{532}$=0.3 for thirty-two canonical cases[2] |
| **Vertical distribution of spectral aerosol extinction coefficient** | Normalized spectral aerosol extinction coefficient of two built-in RRTMG aerosol types[1] is multiplied by $AOD_{532}$=0.3 for thirty-two canonical cases[2] |
| **Atmospheric Composition and Weather** | Assumed constant[3] |
| **Ocean Surface BRDF** | Cox-Munk parametrization (Cox and Munk, 1954; Jin et al., 2011) with a fixed chlorophyl concentration of 0.2 g/m³ |

**Table A1: Theoretical $DARE_T$ calculations for aerosols above clouds in our study and their respective inputs. (1) see**
**"Continental average" and "Urban" aerosol types on Fig. A1; (2) see upper panels (a)-(d) on Fig. A2; (3) $CO_2$, $N_2O$, $CH_4$,**
**$O_2$ and ocean surface wind speed are assumed equal to a single value (i.e., respectively 400 ppmv, 0.3 ppmv, 1.7 ppmv, 0.0**
**kg m³ and 4 m s⁻¹); the pressure, temperature, air density, water vapor and ozone profiles are also assumed constant and**
**illustrated in Table A2; The instantaneous $DARE_T$ uses the Solar Zenith Angle (SZA) at 15ºS latitude and 8ºE longitude on**
**15 September 2016. We then compute twenty-four instantaneous $DARE_T$ values based on twenty-four SZAs (every hour)**
**throughout the day (at the same location and date) and average all instantaneous $DARE_T$ to obtain the diurnal mean $DARE_T$**
**values.**




| Z (km) | P (mb) | T (k) | Air Density | $H_2O$ (g m$^{-3}$) | $O_3$ (g m$^{-3}$) |
|---|---|---|---|---|---|
| 50 | 7.98E-01 | 270.6 | 1.03E+00 | 1.20E-05 | 4.00E-06 |
| … | … | … | … | … | … |
| 15 | 1.21E+02 | 216.6 | 1.95E+02 | 7.20E-04 | 2.10E-04 |
| 14 | 1.42E+02 | 216.6 | 2.28E+02 | 8.40E-04 | 1.90E-04 |
| 13 | 1.66E+02 | 216.6 | 2.67E+02 | 1.80E-03 | 1.70E-04 |
| 12 | 1.94E+02 | 216.6 | 3.12E+02 | 3.70E-03 | 1.60E-04 |
| 11 | 2.27E+02 | 216.8 | 3.65E+02 | 8.20E-03 | 1.30E-04 |
| 10 | 2.65E+02 | 223.2 | 4.14E+02 | 1.80E-02 | 9.00E-05 |
| 9 | 3.08E+02 | 229.7 | 4.67E+02 | 4.60E-02 | 7.10E-05 |
| 8 | 3.57E+02 | 236.2 | 5.26E+02 | 1.20E-01 | 5.20E-05 |
| 7 | 4.11E+02 | 242.7 | 5.90E+02 | 2.10E-01 | 4.80E-05 |
| 6 | 4.72E+02 | 249.2 | 6.60E+02 | 3.80E-01 | 4.50E-05 |
| 5 | 5.41E+02 | 255.7 | 7.36E+02 | 6.40E-01 | 4.50E-05 |
| 4 | 6.17E+02 | 262.2 | 8.19E+02 | 1.10E+00 | 4.60E-05 |
| 3 | 7.01E+02 | 268.7 | 9.09E+02 | 1.80E+00 | 5.00E-05 |
| 2 | 7.95E+02 | 275.1 | 1.01E+03 | 2.90E+00 | 5.40E-05 |
| 1 | 8.99E+02 | 281.6 | 1.11E+03 | 4.20E+00 | 5.40E-05 |
| 0 | 1.01E+03 | 288.1 | 1.23E+03 | 5.90E+00 | 5.40E-05 |

**Table A2. Atmospheric profiles of pressure, temperature, air density, water vapor and ozone used in the calculation of**
**$DARE_T$ (see Table A1 and legend of Table A1 for constant $CO_2$, $N_2O$, $CH_4$, $O_2$ and ocean surface wind speed values).**






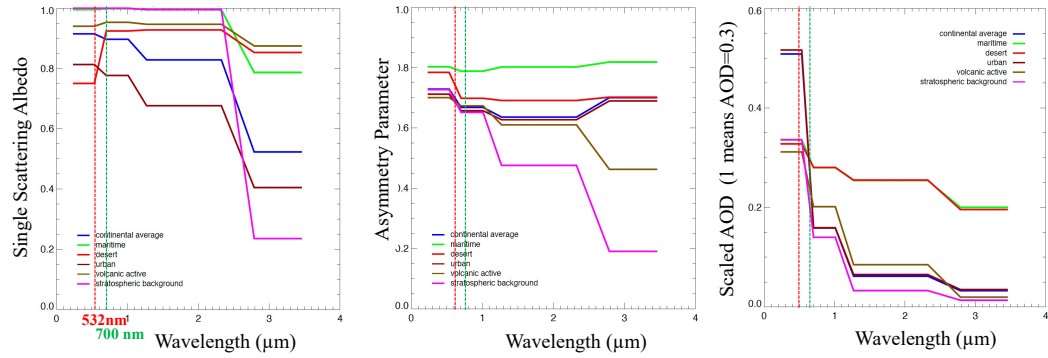

| RRTMG Aerosol Type | [SSA, ASY] at 532 nm | [SSA, ASY] at 700 nm |
|---|---|---|
| Continental Average | [0.92, 0.72] | [0.90, 0.67] |
| Urban | [0.82, 0.70] | [0.78, 0.65] |
| Stratospheric Background | [1.00, 0.72] | [1.00, 0.65] |




**Figure A1. RRTMG "build-in" aerosol types used in the calculation of DARE_T (see Table A1). SSA is 0.92 (0.90), 0.82 (0.78),**
**1.00 (1.00) for RRTMG "Continental Average", "Urban" and "Stratospheric Background" aerosol types at 532 (700) nm.**
**Note that RRTMG "Continental Average" seems to correspond roughly to biomass burning smoke aerosol types in Russell**
**et al. (2014). Also note that RRTMG "Urban" seems to correspond to aerosols with considerably higher light absorption**
**properties than the smoke types in Russell et al. (2014). ASY is 0.72 (0.67), 0.70 (0.65) and 0.72 (0.65) for RRTMG**
**"Continental Average", "Urban" and "Stratospheric Background" at 532 (700) nm. Aerosol types are taken from the**
**Optical Properties of Aerosols and Clouds (OPAC) software [Hess et al., 1998].**







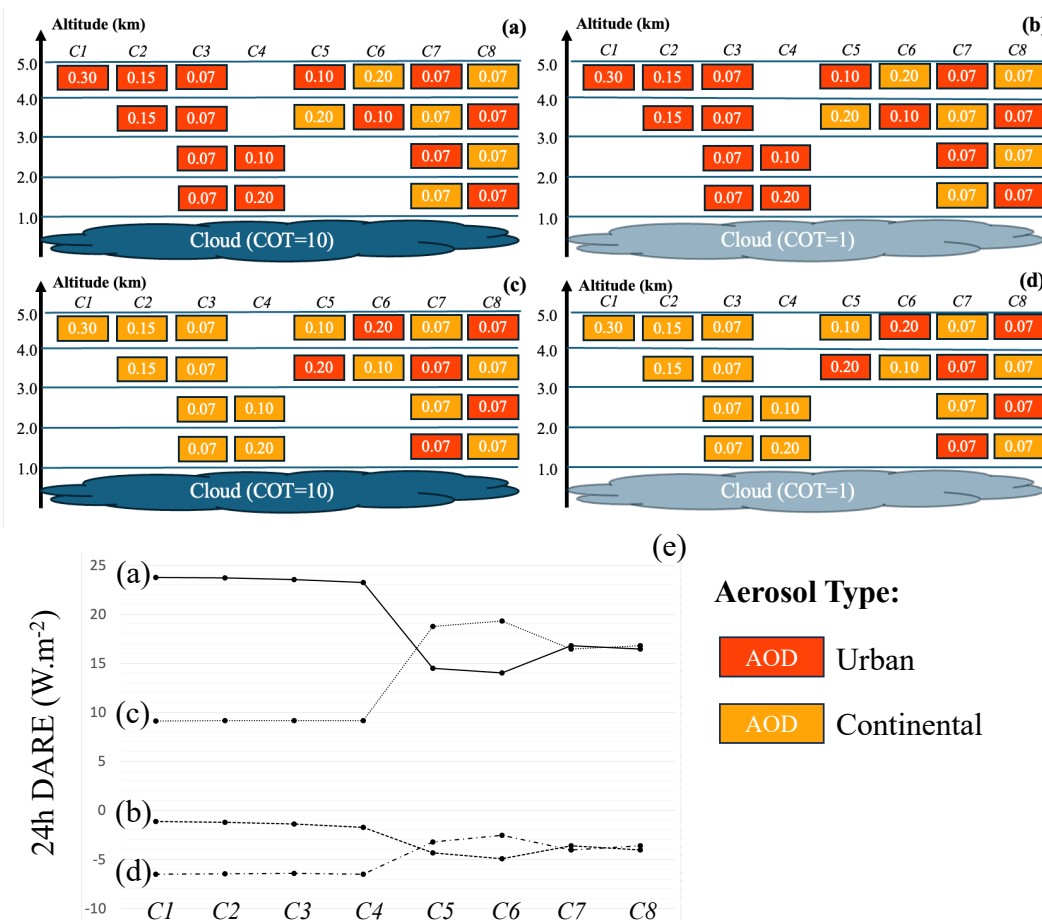



**Figure A2. Diurnal mean theoretical DARE$_T$ (in W·m$^{-2}$) results in (e) for thirty-two canonical cases (i.e., eight cases in (a),**
**(b), (c) and (d)) where we vary COT, the number of aerosol layers over clouds, the order of aerosol types and the loading**
**of aerosols over clouds. Orange and red boxes depict two "build-in" RRTMG aerosol types, respectively "Continental**
**average" in orange and "Urban" in red; see Fig. A1 for the optical and microphysical properties of these aerosol types. The**
**vertical distribution of spectral SSA and extinction coefficient are weighed by the AOD above clouds that is assumed**
**constant and equal to 0.3 at 532nm (i.e., in the 442-625 nm RRTMG broadband channel). See Table A1 for a list of the**
**inputs to the DARE$_T$ calculations.**






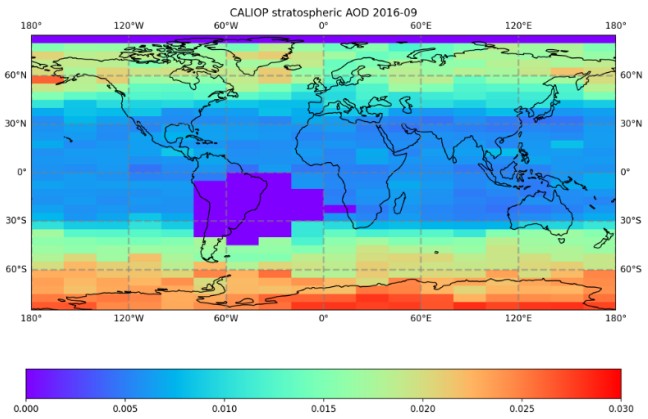


**Figure A3: Stratospheric aerosols that are deleted when computing DARE$_S$ (see Table 2)**





| (1) Method to compute DARE$_S$ in each atmospheric scenario[i] | | | | |
|---|---|---|---|---|
| Atmospheric Scenario | S1 | S2 | S3 | S4 |
| Aerosol Properties | CALIOP$_{ACAOD\_DR}$ (V1, V2 or V3) | CALIOP$_{ACAOD\_standard}$ (V1, V2 or V3) | | CALIOP$_{ODAOD}$ when valid; if CALIOP$_{ODAOD}$ not valid, CALIOP$_{AOD\_standard}$ (V1, V2 or V3) |
| | MERRA-2 composition above clouds; MERRA-2 AOD, composition, ATH and ABH below clouds*[i] | | | MERRA-2 composition in clear skies[i] |
| Cloud Properties | MODIS$_{Cloud}$ CWP and CER; CALIOP CTH; CBH = CTH – 500m | | | N/A |
| (i)* | If MERRA extinction < 0.014 km-1, assume no aerosols; aerosol composition is informed by spectral vertical SSA, ASY and extinction | | | |
| (2) More information on CALIOP aerosol parameters: | | | | |
| CALIOP$_{ACAOD\_DR}$ and corresponding ATH and ABH | Median value of single shot CALIOP$_{ACAOD\_DR}$ from Hu et al. [2007] using Column_Particulate_Optical_Depth_Above_Opaque_Water_Cloud_532 in CAL_LID_L2_05kmMLay product within 5km that is including the 1km stretch; no filters or QA flags (e.g., extinction flag) on CALIOP$_{ACAOD\_DR}$ at the time of writing; if extinction corresponds to < 0.07km$^{-1}$ [Rogers et al., 2011], assume no aerosols*; Stratospheric Optical Depth (SOD) is removed from each profile[ii]; ATH= CALIOP$_{vfm}$ uppermost ATH[iii]; ABH=CTH | | | |
| CALIOP$_{ACAOD\_standard}$ and corresponding ATH and ABH | Integration of extinction profile between uppermost aerosol layer and cloud top height using Extinction_Coefficient_532 in CALIOP 5km aerosol profile product; Extinction flag for CALIOP$_{ACAOD\_standard}$ needs to be 0,1,2; if extinction < 0.07km$^{-1}$ [Rogers et al., 2011], assume no aerosols; ATH= CALIOP$_{vfm}$ uppermost ATH[iii]; ABH=CTH | | | |
| CALIOP$_{AOD\_standard}$ and corresponding ATH and ABH | Integration of extinction profile between uppermost aerosol layer and ocean surface using Extinction_Coefficient_532 in CALIOP 5km aerosol profile product; Extinction flag for CALIOP$_{AOD\_standard}$ needs to be 0,1,2; if extinction < 0.07km-1 [Rogers et al., 2011], assume no aerosols; ATH = CALIOP$_{vfm}$ uppermost ATH[iii]; ABH = CALIOP$_{vfm}$ lowermost ABH | | | |



| CALIOP$_{ODAOD}$ and corresponding ATH and ABH | We use Ocean Derived Column Optical Depths (ODCOD) from Venkata and Reagan (2016) and Ryan, (2024) (i.e., ODCOD_Effective_Optical_Depth_532 in CAL_LID_L2_05kmMLay product); (0) single shot surface IAB 532 < 0.0413 and surface integrated depolarization ratio < 0.05; (1) no clouds detected at 1km or at SS; (2) if 3< wind <15m.s-1, then use median of all single shot CALIOP$_{ODCOD}$ within 5km that includes 1 km stretch; no official filters or QA flags (e.g., extinction flag) on CALIOPODCOD at the time of writing; Stratospheric Optical Depth (SOD) is removed from each profile[ii]; ATH = CALIOP$_{vfm}$ uppermost ATH[iii]; ABH = CALIOP$_{vfm}$ lowermost ABH |
|---|---|
| CALIOP$_{ODAOD}$ or CALIOP$_{AOD\_standard}$? | We start with CALIOP$_{ODAOD}$; If conditions are not met for (0), (1) and (2) in the line above, then we use CALIOP$_{AOD\_standard}$ |
| (ii) | We compute a zonal SOD from the equal-angle data product, then interpolate the zonal data to the latitude grid of the CALIPSO granule observations. Then we remove the SOD from CALIOP$_{ACAOD\_DR}$ |
| (iii)* | ATH extension: If there is (1) no valid CALIOP$_{vfm}$ uppermost ATH corresponding to a valid ACAOD for S1-S4 and (2) a valid median ATH ±10km centred on the invalid ATH then ATH is replaced by ±10km median ATH; if (1) but not (2), then ATH=median(orbit section); |
| **(3) Three versions of CALIOP-derived AOD** | |
| V1* | Consider only valid CALIOP$_{ACAOD\_DR}$ paired with ATH$_{ACAOD\_DR}$, ABH$_{ACAOD\_DR}$ and CALIOP$_{ACAOD\_standard}$ paired with CALIOP$_{ACAOD\_standard}$; when there is no valid CALIOP$_{ACAOD\_DR}$ or CALIOP$_{ACAOD\_standard}$ data, do not replace |
| V2* | For each 1km stretch, if CALIOP$_{ACAOD\_DR}$ (or CALIOP$_{ACAOD\_standard}$) is not valid for S1, S2 or S3, invalid point is replaced by ±10km median single shot CALIOP$_{ACAOD\_DR}$; For S3, if ±10km median single shot CALIOP$_{ACAOD\_DR}$ is still not available, invalid point is replaced by 5km CALIOP$_{ACAOD\_Standard}$ |
| V3* | For each 1km stretch, median of rolling ±10km median of single shot CALIOP$_{ACAOD\_DR}$ for S1, S2, S3. If the latter does not exist for S3, then use 5km CALIOP$_{ACAOD\_standard}$; For each 1km stretch, ATH is replaced everywhere by rolling ±10km median V1 ATH; If there is no rolling median ATH available, ATH=median(orbit section) |

**Table A3: (1) Detailed description of aerosol and cloud property inputs to DARE$_S$ calculations for each atmospheric**
**scenario, (2) more information on CALIOP-derived input aerosol parameters and (3) description of three CALIOP-derived**
**AOD versions. The asterisks denote where we have assessed the effects of modifying the parametrization in the calculation**
**of DARE$_S$. All these effects are summarized in section 2.1.4. We have selected to display DARE$_S$ results corresponding to**
**version 2 in the main sections of this paper and for our DARE$_S$ algorithm moving forward.**




| | | Number, Averaged Instantaneous and 24h DAREs | | | | | | | | Effects of: | | | | | |
|---|---|---|---|---|---|---|---|---|---|---|---|---|---|---|---|
| **Threshold on extinction?** | | no | yes | yes | yes | yes | yes | yes | yes | Adding a threshold on extinction coefficient | Adding aerosol below clouds | Extending aerosol top height | Using AOD V2 compared to AOD V1 | Using AOD V2 compared to AOD V3 | Using clouds corrected for aerosols above |
| **ATH extended?** | | yes | yes | yes | no | yes | yes | yes | yes | | | | | | |
| **AOD version?** | | V1 | V1 | V1 | V1 | V2 | V3 | V2 | V2 | | | | | | |
| **Aerosol below clouds?** | | no | no | yes | yes | yes | yes | yes | yes | | | | | | |
| **Clouds corrected for aerosols above?** | | no | no | no | no | no | no | no | yes | | | | | | |
| **AOD?** | | ≥0 | | | | | | > 0.3 | | | | | | | |
| **Where?** | | All-sky (S1-S4) | | | | | | Thick clouds (S1) | | All-sky (S1-S4) | | | | | Thick clouds (S1) |
| **Number** | **9/18/16** | 1107 | 1107 | 1107 | 1097 | 1160 | 1160 | 154 | 154 | 0 | 0 | 10 | 53 | 0 | 0 |
| | **9/20/16** | 878 | 878 | 878 | 878 | 887 | 887 | 597 | 597 | 0 | 0 | 0 | 9 | 0 | 0 |
| | **8/13/17** | 741 | 741 | 741 | 739 | 806 | 806 | 23 | 23 | 0 | 0 | 2 | 65 | 0 | 0 |
| **DAREs instant** | **9/18/16** | 23.2 | 22.4 | 21.4 | 21.7 | 20.3 | 20.3 | 38.3 | 37.3 | 0.8 | 1.0 | 0.3 | 1.1 | 2.3 | 1.9 |
| | **9/20/16** | 35.2 | 35.0 | 34.5 | 34.5 | 34.2 | 34.2 | 46.9 | 48.3 | 0.2 | 0.5 | 0.0 | 0.4 | 2.9 | 4.1 |
| | **8/13/17** | 12.4 | 12.4 | 10.8 | 10.8 | 9.5 | 8.6 | 80.7 | 82.5 | 0.0 | 1.6 | 0.1 | 1.3 | 3.2 | 3.0 |
| | | Ai | Bi | Ci | Di | Ei | Fi | Gi | Hi | mean(\|Ai-Bi\|) | mean(\|Bi-Ci\|) | \|mean(Ci)-mean(Di)\| | \|mean(Ci)-mean(Ei)\| | mean(\|Ei-Fi\|) | mean(\|Gi-Hi\|) |






**Table A4: Effects of (i) adding a lower threshold on CALIOP and MERRA-2 extinction coefficients, (ii) adding MERRA-2**
**aerosol below clouds, (iii) extending aerosol Top Height (ATH) when there is no valid ATH from the CALIOP standard**
**product, (iv) using AOD V2 instead of V1, (v) using AOD V3 instead of V2 and (vi) using clouds corrected for aerosol above**
**when AOD>0.3. Latitudes are selected between 6ºS and 20ºS.**





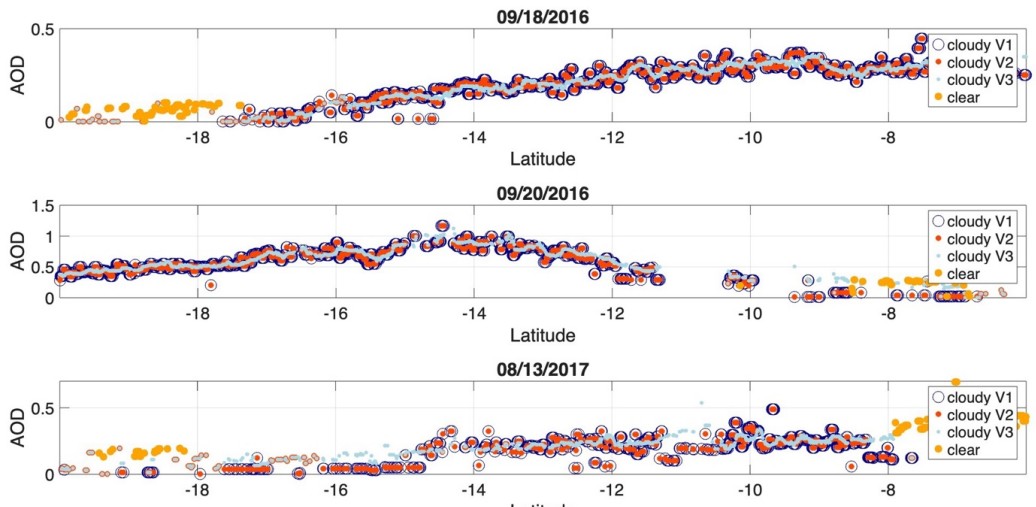

**Figure A4: Evolution of AOD V1, V2, and V3 above clouds (see Table A2 for definition of these versions) and AOD in clear**
**skies for our three case studies. We eventually select AOD V2 in this paper. Latitudes are selected between 6ºS and 20ºS.**




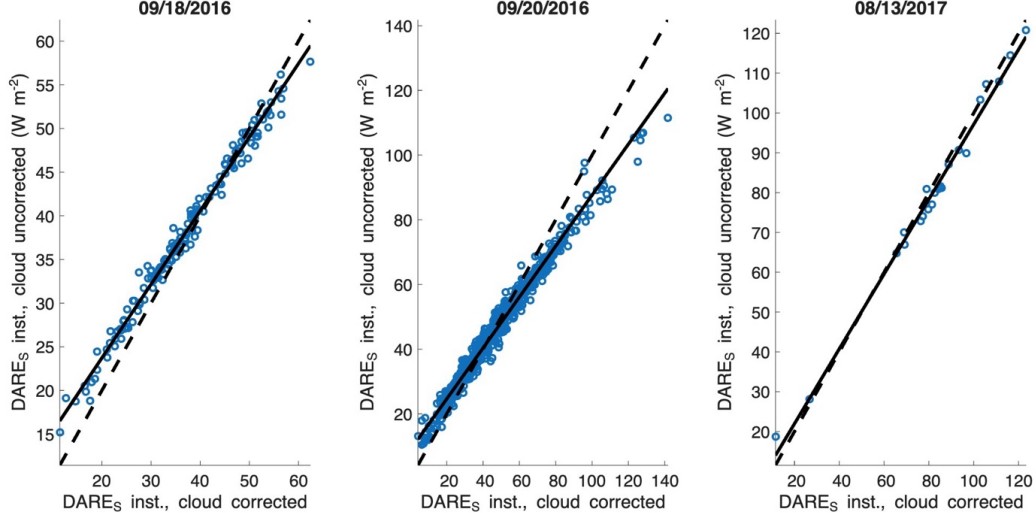


**Figure A5: Semi-observational instantaneous cloudy DARE$_S$ (W·m$^{-2}$) (see Table 2) using MODIS COT corrected for aerosol**
**above (x-axis) vs. MODIS COT uncorrected for aerosol above (y-axis). We show only values with AOD>0.3 above clouds.**
**See table A5 for linear regression and correlation statistics. Latitudes are selected between 6ºN and 20ºS**






| | | 9/18/16 | 9/20/16 | 8/13/17 |
|---|---|---|---|---|
| Mean cloudy DARE$_S$ instantaneous with clouds corrected | | 37.32 | 48.29 | 82.46 |
| Mean cloudy DARE$_S$ instantaneous with clouds uncorrected | | 38.33 | 46.91 | 80.66 |
| DARE$_S$ instantaneous with clouds corrected vs. clouds uncorrected for aerosol above | R$^2$ | 0.99 | 0.98 | 0.99 |
| | Slope, Offset | 0.84, 6.88 | 0.79, 8.87 | 0.94, 3.15 |
| | N | 154 | 597 | 23 |
| | RMSE | 2.3 | 5.66 | 3.41 |
| | Difference of Mean | 1 | 1.38 | 1.8 |
| | Mean of Difference | 1.86 | 4.06 | 2.98 |
| Mean COT with clouds corrected | | 15.5 | 9.71 | 35.56 |
| Mean COT with clouds uncorrected | | 13.64 | 8.11 | 28.35 |
| COT with clouds corrected vs. clouds uncorrected for aerosol above | R$^2$ | 0.98 | 0.95 | 0.95 |
| | Slope, Offset | 0.72, 2.44 | 0.60, 2.29 | 0.69, 3.88 |
| | N | 154 | 597 | 23 |
| | RMSE | 2.45 | 2.2 | 8.55 |
| | Difference of Mean | 1.86 | 1.61 | 7.2 |
| | Mean of Difference | 1.88 | 1.63 | 7.24 |

**Table A5: Statistics behind figure A5 – Comparison between DARE$_S$ instantaneous or COT with clouds corrected vs. clouds**
**uncorrected for aerosol above**




**Results**

| Averaged Values | 9/18/16 | | | 9/20/16 | | | 8/13/17 | | |
|---|---|---|---|---|---|---|---|---|---|
| | **S1** | **S2** | **S3** | **S1** | **S2** | **S3** | **S1** | **S2** | **S3** |
| **Number** | 968 | 31 | 68 | 724 | 22 | 94 | 334 | 64 | 221 |
| **DARE 24h** | 10.33 | 2.03 | -1.37 | 17.62 | 11.98 | -0.13 | 14.11 | 6.6 | -1.2 |
| **DARE 24h Uncertainty** | 0.1 | 0.3 | 0.08 | 0.14 | 0.72 | 0.14 | 0.27 | 0.3 | 0.06 |
| **DARE Instant** | 24.99 | 3.65 | -4.1 | 42.29 | 27.03 | -1.98 | 35.91 | 14.39 | -4.7 |
| **DARE Instant Uncertainty** | 0.26 | 0.72 | 0.21 | 0.39 | 1.91 | 0.39 | 0.73 | 0.77 | 0.15 |
| **COT** | 11.6 | 5.95 | 1.82 | 7.64 | 6.05 | 2.14 | 14.62 | 6.01 | 1.59 |
| **CWP** | 86.42 | 38.45 | 11.65 | 40.71 | 31.86 | 14 | 87.24 | 30.95 | 11.05 |
| **CWP Uncertainty** | 0.416 | 2.912 | 3.855 | 0.548 | 3.425 | 2.094 | 0.837 | 1.939 | 2.608 |
| **CER** | 11.43 | 10.1 | 10.32 | 8.27 | 8.28 | 10.75 | 8.71 | 8.11 | 12.42 |
| **CALIOP_CF** | 1 | 0.98 | 0.92 | 1 | 0.98 | 0.9 | 1 | 0.97 | 0.92 |
| **MODIS_CF** | 1 | 1 | 0.82 | 1 | 1 | 0.88 | 1 | 1 | 0.8 |
| **AOD above Clouds** | 0.24 | 0.11 | 0.06 | 0.63 | 0.53 | 0.19 | 0.25 | 0.22 | 0.1 |
| **AOD Uncertainty** | 0.002 | 0.011 | 0.006 | 0.004 | 0.024 | 0.012 | 0.004 | 0.008 | 0.002 |
| **SSA at highest altitude** | 0.84 | 0.82 | 0.81 | 0.87 | 0.86 | 0.88 | 0.81 | 0.8 | 0.83 |
| **ASY at highest altitude** | 0.64 | 0.63 | 0.64 | 0.67 | 0.66 | 0.68 | 0.6 | 0.59 | 0.62 |
| **EAE at highest altitude** | 1.86 | 1.77 | 1.67 | 1.89 | 1.91 | 1.81 | 2.1 | 2.1 | 1.92 |
| **ATH** | 4.59 | 4.38 | 4.44 | 5.24 | 4.98 | 4.26 | 2.95 | 3.02 | 2.7 |
| **CTH** | 1.02 | 0.93 | 0.86 | 0.64 | 0.59 | 0.76 | 1.11 | 0.91 | 0.73 |

**Table A6: Averaged aerosol, cloud and DARE properties per atmospheric scenario, and case study. We display DARE$_S$**
**results corresponding to version 2 in the main sections of this paper and for our DARE$_S$ algorithm moving forward. SSA**
**uncertainty is fixed at 0.05 and ASY uncertainty is fixed at 0.02 (see Table 4). Latitudes are selected between 6ºS and 20ºS.**





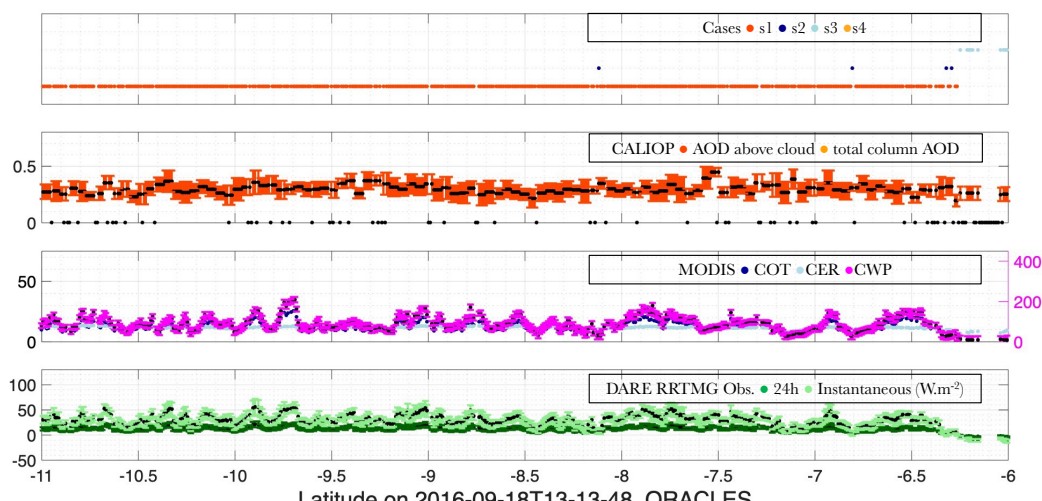



**Figure A6: Key input parameters to our DARE$_S$ calculations, together with the DARE$_S$ values themselves (diurnal mean**
**and instantaneous) along the CALIOP track on 09/18/2016. From the top to the bottom panel -- S1, S2, S3 and S4 cases, the**
**V2 AOD, COT, CER and CWP. Cloud retrieved optical properties are not corrected for aerosols above them. Instead of**
**showing latitudes between 6ºS and 20ºS, we reduce the latitude range here from 6ºS to 11ºS for visibility.**




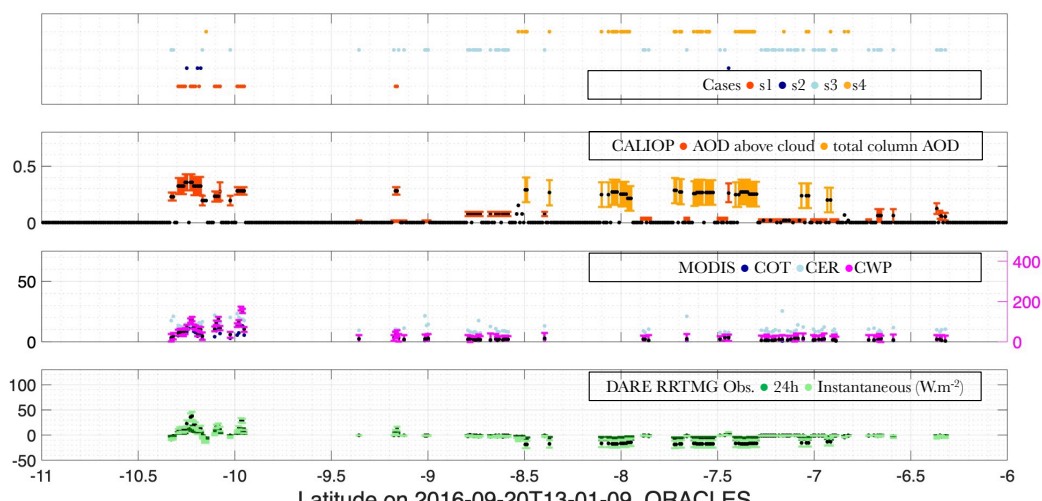



**Figure A7: See Fig. A6 but for 09/20/2016.**








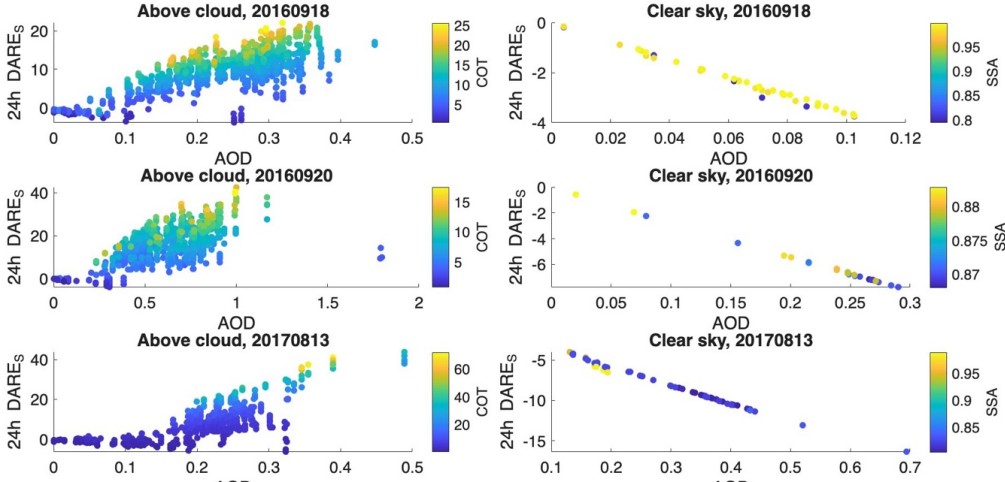


**Figure A8: DARE in clear skies as a function of AOD and SSA (top row) and DARE above clouds as a function of AOD and COT (bottom row) on 09/18/2016, 09/20/2016 and 08/13/2017. Cloud retrieved optical properties are not corrected for aerosols above them. Latitudes are selected between 6ºS and 20ºS.**







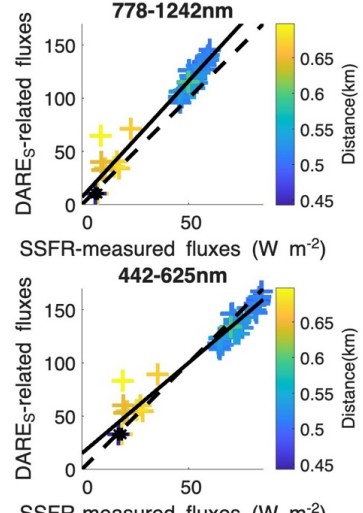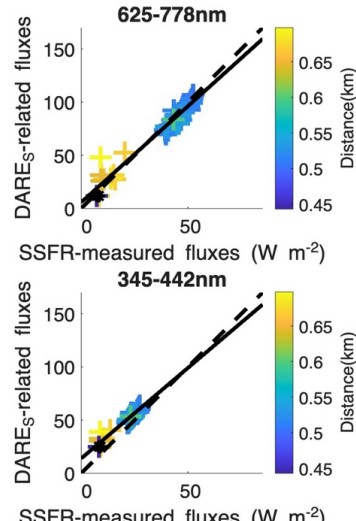




**Figure A9: SSFR-measured fluxes vs. DARE$_S$-related fluxes (W·m$^{-2}$) in four RRTMG broadband channels. Points are**
**colored by distance between the aircraft and the CALIOP track in km. Black stars are points in clear-sky conditions (S4).**
**See second part of Table 6 in the text for statistics.**

