# Peer review of "All-Sky Direct Aerosol Radiative Effects Estimated from"

_EGUsphere, 2025_

## Author Response (AR1)

We are grateful for both referees' thoughtful reviews as addressing them has strengthened our paper. We have addressed all of them carefully as described below.
Kindest regards,
Meloë Kacenelenbogen on behalf of all co-authors.

**Referee #1**
*This manuscript presents a novel study aimed at developing improved satellite-derived estimates of Direct Aerosol Radiative Effects (DARE). This work introduces a new method for computing DARE by leveraging satellite sensors CALIOP and MODIS combined with MERRA-2 vertical distributions of aerosol properties and atmospheric conditions. With the advantage of active remote sensing, this approach is capable of performing all-sky retrievals of SW aerosol radiative effects useful for comparisons with dedicated field studies such as the ORACLES aircraft campaign. Overall, this manuscript is well-written, and the figures are clearly presented. Below are a few suggestions that may help further improve the manuscript:*

*Major Comments:*
- *The current notation for the DARE calculations (i.e., $DARE_s$ and $DARE_p$, as defined) are challenging to read. These subscripts are fairly difficult to distinguish within the text. To enhance clarity and reader comprehension, it could be helpful to explore alternative naming conventions. For example, labeling them DARE_obs (for the method utilizing observations) and DARE_param (for the parameterized method) might be more intuitive.*

We have changed $DARE_s$ into DARE_obs, $DARE_p$ into DARE_param and $DARE_T$ into DARE_theo throughout the manuscript (includes text, tables and figures)

- *As the paper describes very well, the Southeast Atlantic presents an excellent natural laboratory for studying DARE and validating against ORACLES data, particularly given the variable atmospheric conditions. To strengthen the broader significance of this work, it would be helpful to elaborate on the potential for extending the findings beyond this region. Specifically, a discussion of the factors that might influence the transferability of these results to other parts of the globe would be particularly insightful.*

In the discussion section, it now reads:
"We plan to apply our DARE_obs calculations to multiple years of combined satellite and model data over different regions of the world. The most important factors influencing the transferability of our method to regions of the globe outside the Southeast Atlantic are (i) different Earth's surfaces (i.e., ocean vs. different land types) and (ii) different horizontal, vertical and temporal distributions of aerosol and cloud types and amounts. Our method requires aerosols in cloud-free skies, and above and below single thick, thin and/ or broken low warm liquid clouds. Kacenelenbogen et al. (2019) define six major global aerosol "hotspots" over single thick low warm liquid clouds (i.e., different aerosol regimes above the same type of clouds) in the northeast Pacific, southeast Pacific, tropical Atlantic, southeast Atlantic, Indian ocean, offshore from western Australia and northwest Pacific (see their Fig. 6; and Table 2 for a list of studies over these regions). According to Fig. 7d of Kacenelenbogen et al. (2019), the region of Southeast Atlantic (this paper) shows the highest mean annual percentage of high AOD values above clouds compared to the five other regions. Note that we also plan to apply our DARE_obs calculations to regions that show different cloud regimes in addition to different aerosol regimes (e.g., the Southeast Atlantic, the tropical Atlantic, and a region encompassing the latter two representing the transition between these two regimes). We then plan to use this larger DARE_obs dataset for

different atmospheric scenarios, over specific regions of the world and linked to key cloud, aerosol and surface input parameters to assess the order of importance of these parameters in DARE_obs calculations for specific aerosol and cloud regimes."

*Minor Comments:*
- *Lines 178-195: While well-structured and informative, this paragraph is difficult to read. I would recommend shorter sentences with fewer em dashes to improve clarity.*
- *Line 188: It can be quite challenging to distinguish a "~' and a "-" in the text. As an alternative, " ~-7 to ~-1" can be rewritten as "approximately -7 to -1".*

We modified the manuscript accordingly. It now reads:

"Like Table 2 for DARE_obs and DARE_param, Table A1 in the appendix lists the input parameters to our DARE_theo calculations. DARE_theo is computed for two types of single low warm liquid clouds (i.e., COT=1, CER=12 and CWP=8 vs. COT=10, CER=12 and CWP=80) and varying vertical distributions of RRTMG "build-in" aerosol types (see Fig. A1) while keeping cloud heights, AOD, ASY, atmospheric composition, weather and ocean surface BRDF constant (see thirty-two canonical cases illustrated in panels a, b, c, and d of Fig. A2 where we vary the order and amount of two aerosol types over clouds in the vertical). No matter which type and which vertical distribution of aerosol above cloud is considered, DARE_theo values are lower when aerosols are present above a cloud of COT equal to 1 (cases (e-b) and (e-d)), compared to a COT equal to 10 (cases (e-a) and (e-c) in Fig. A2). This is illustrated by changes of approximatively -7 to -1 $W \cdot m^{-2}$ for (e-b) and (e-d) vs. approximatively 9 to 24 $W \cdot m^{-2}$ for (e-a) and (e-c) of Fig. A2. We also record lower DARE_theo values when adding more scattering aerosols (i.e., "continental" aerosol type) to already absorbing aerosols (i.e., "urban" aerosol type). In effect, DARE_theo values drop from approximatively 24 to 14 $W \cdot m^{-2}$ when aerosols are more scattering above a cloud of COT equal 10 (see C1-C4 in (e-a) vs. C5-C8 in (e-a) of Fig. A2). And DARE_theo values drop from approximatively -1 to -5 $W \cdot m^{-2}$ when aerosols are more scattering above a cloud of COT equal 1 (see C1-C4 in (e-b) vs. C5-C8 in (e-b) of Fig. A2). In conclusion, the variability of these DARE_theo calculations confirm, as expected, that our semi-observational DARE_obs calculations need to account for the vertical order and location of aerosol types and aerosol amount."

- *Figure 5: The legend currently needs improvement to enhance readability. The overlay of data points on the legend makes it difficult to discern the labels. Additionally, the initial word of each legend (e.g. Cases, CALIOP, MODIS, etc.) are distracting and could be revised for improved clarity.*

We modified figure 5, 7, A6 and A7 accordingly. Legends are now clarified and visible.

- *Figure 6: The near 1:1 agreement in DARE calculations observed on 8/13/2017, suggests atmospheric conditions that are notably different from the two preceding cases. This is an interesting finding. Could you elaborate on the specific atmospheric conditions that might explain this unique 8/13/2017 scenario within the figure's discussion?*

The paper now reads:

"When evaluating our semi-observational DARE_obs with coincident parametrized DARE_param over all types of clouds (i.e., S1, S2 and S3 in Table 3) and for our three case studies, we find a generally satisfying agreement ($R^2$=0.87 to 0.99, slope=0.80 to 0.99, offset =0.37 to 8.30, N=619 to 1067 in (1) Table 6). We posit that the slight differences between DARE_obs and DARE_param (see, for example, the mean cloudy DARE_param and DARE_obs values in panel (1) of Table 6)

pertain to how they are computed. On the one hand, we assume MERRA-2's vertical distribution of SSA for the DARE_obs calculations, even though the SSA magnitude lies outside the observed SSA variability during ORACLES (i.e., as seen in Fig. 4b in Cochrane et al. (2021), the peak of the *in-situ* SSA values measured at 532 nm is between 0.85 and 0.86). By invoking this assumption, we can either overestimate DARE_obs if the MERRA-2 SSA value is too low or underestimate DARE_obs if the MERRA-2 SSA value is too high. For example, when computing DARE_theo (see Fig. A2), we record lower DARE_theo values (by ~10 W m$^{-2}$) when adding more scattering aerosols (i.e., "continental") to already absorbing aerosols (i.e., "urban") over a thick cloud (COT=10). A second example is seen on 09/20/2016, where the two data points showing high AOD values above clouds (in yellow) and causing an offset in the DARE_param vs. DARE_obs regression line (~8 in Table 6) are likely due to an underestimation of MERRA-2 SSA, which in turn causes an overestimation of DARE_obs compared to DARE_param. On the other hand, while DARE_param is computed using the same AOD and cloud microphysical properties as DARE_obs, the DARE_param framework was developed specifically for aerosols above homogeneous cloud conditions (i.e., S1) and thus might not apply as well to broken and/ or thin clouds (i.e., S2 and S3). The various amounts of S1, S2 and S3 cases during our three case studies (illustrated in Fig. 3) likely influence the DARE_param accuracy. We also note a distinctive feature in Fig. 6 on 09/18/2016 away from the 1:1 line for low AOD and CALIOP cloud fractions below 1 (black crosses). This feature is very likely due to cloud inhomogeneities paired with low AOD values"

- *Figure 7: The bottom subplot displays flux difference values, but some data points appear to extend beyond the figure's axes. Are these outlying data points of lesser significance to the overall analysis?*

We've added this to the text:

"For increased visibility and because the spatial satellite-aircraft colocation is deteriorated from ~9.2ºS to 7.9ºS in latitude (and hence the data is of lesser significance to the overall analysis), we allow a few data points in panel (e) to extend beyond the figure's axes"

**Referee #2**
**Questions and Comments**
1. *Clarity of CALIOP AOD Product Usage in DARES: Table 2 mentions CALIOP AOD at 532nm is a combination of CALIOP ACAOD_standard, CALIOP ACAOD_DR, CALIOP AOD_standard, and CALIOP ODAOD. The paper details these products, but the specific logic or conditions under which each is chosen or how they are "combined" for a single AOD input to DARES could be more explicit in this section.*

We've added:

(1) In section 2.1.1:

"Table A3 describes how CALIOP AOD is chosen to be equal to CALIOP$_{ACAOD\_standard}$, CALIOP$_{ACAOD\_DR}$, CALIOP$_{AOD\_standard}$ and/ or CALIOP$_{ODAOD}$ (see Table 1) in different atmospheric scenarios (i.e., clear skies, or among thick and/or thin clouds present)."

(2) In Table 2:

"Table A3 describes how CALIOP AOD is chosen to be CALIOP$_{ACAOD\_standard}$, CALIOP$_{ACAOD\_DR}$, CALIOP$_{AOD\_standard}$ and/ or CALIOP$_{ODAOD}$ in different atmospheric scenarios."

2. *DAREP Methodology Details: While DARES is detailed, DAREP (used for evaluation) is only cited. A brief summary of how it contrasts with DARES in handling key parameters*

*(especially clouds and aerosol vertical distribution) might be beneficial for context within this paper.*

We've added:

(1) In the legend of Table 2:

**Two different DARE calculations (i.e., semi-observational DARE_obs described in section 2.1, and parametrized DARE_param described in section 2.2)**

(2) Under Table 2:

"The parametrization that allows us to compute DARE_param is described in section 2.2. It builds on a method that systematically links aircraft observations of SSFR-linked spectral fluxes to aerosol optical thickness and other parameters using nine cases from the 2016 and 2017 ORACLES campaigns. This observationally driven link is expressed by a parametrization of the shortwave broadband DARE in terms of the mid-visible AOD and scene albedo."

3. *MERRA-2 SSA Bias: The paper acknowledges that MERRA-2/GEOS SSA tends to be higher than in-situ measurements (underestimating absorption), particularly for biomass burning aerosols prevalent in the ORACLES study region. The impact of this known bias on the DARES results, especially for positive DARE over clouds, might be important to discuss.*

We've added in section 2.1.1:

"We expect, according to the DARE_theo calculations illustrated in Fig. A2, that a high bias in the MERRA-2 estimated SSA, if not compensated by other factors, would cause a low bias in DARE_obs calculations (see, for example, lower DARE_theo values in (e-a) for C5-C8 where SSA is higher compared to higher DARE_theo values in (e-a) for C1-C4 where SSA is lower)."

4. *Extinction Coefficient Thresholds: The paper sets extinction coefficient thresholds for CALIOP (0.07 km$^{-1}$) and MERRA-2 (0.014 km$^{-1}$) to define "aerosol-free" conditions. The MERRA-2 threshold is scaled from the CALIOP one based on average layer thickness. The sensitivity of DARE results to these threshold choices could be relevant.*

We've clarified Table A3 – It now reads "we have assessed the effects of these factors in the calculation of DARE_obs" and one of them is (E-1) i.e., "Apply threshold on extinction"

We've added in section 2.1.1:

"In section 2.1.4, we demonstrate that adding or removing such a threshold on the aerosol extinction coefficient leads to insignificant differences in mean instant DARE_obs values (up to 0.8 W.m$^{-2}$) for all three case studies."

We've added in section 2.1.4:

"Regarding categories (E-1), (E-2) and (E-3), the effects add up to a small N=10 1km-data points in Table A4 and lead to a small difference in mean instant all-sky (S1-S4) DARE_obs of maximum ~1.6 W·m$^{-2}$."

5. *Regarding Table 2, Footnote (1): Could you clarify the entry "O2 mass density = 0.0 kg m3"? Does this imply that O2 is not a variable input in the radiative transfer model, being part of the standard atmospheric profile, rather than having zero density?*

We have added in the legend of Table 2:

"$O_{2\text{ mass density}}$, which is also a required input to RRTMG, is assumed to be 0.0 kg m$^{3}$."

6. *Corrected vs. Uncorrected MODIS Cloud Properties: Section 2.1.1 mentions that using corrected MODIS cloud properties (accounting for above-cloud aerosols) can sometimes worsen agreement with other CER measurements depending on the spectral channel. Which version (corrected or uncorrected) is predominantly used for the DARES results presented for the ORACLES case studies, and what was the rationale for this choice?*

We've added in section 2.1.1:

"In section 2.1.4, we demonstrate that correcting cloud properties for aerosol above them leads to insignificant differences in mean instant DARE_obs values (up to 4 $W.m^{-2}$) for all three case studies."

We've also added in section 2.1.1:

"In the end, the effects of (E-1) through (E-6) all lead to small differences in DARE_obs below a threshold of 6 $W.m^{-2}$, which represents the accuracy of total fluxes in overcast conditions when comparing RRTMG-SW with other radiative transfer schemes (such as RRTM-SW)."

7. *Aerosol Information Below Clouds: For aerosols below clouds, MERRA-2 is used for extensive and intensive properties. How are situations handled where CALIOP detects clouds, but MERRA-2 shows no significant aerosol below them, or vice-versa? Is there a priority system or a check for consistency?*

We've added in section 2.1.2:

"As described in Table 2, on the one hand, MODIS and CALIOP satellites are used to detect and characterize clouds, define aerosol height, and provide aerosol extinction coefficients above clouds and in non-cloudy skies. MERRA-2, on the other hand, is used to define aerosol top and base heights below clouds and provide the vertical distribution of spectral ASY, SSA, and extinction coefficient above, below clouds and in non-cloudy skies, along with information about atmospheric composition, weather, and ocean surface winds. We emphasize that we use MERRA-2 aerosol and atmospheric data regardless of any MERRA-2 simulated clouds (i.e., we do not use MERRA-2 cloud simulations in any way), nor do we assess cloud agreement between MERRA-2 and satellite observations in this paper."

8. *Definition of "Thick, Thin, and/or Broken" Clouds: Section 2.1.3 (referenced in Table 2) will describe atmospheric scenarios including "thick, thin and/ or broken liquid cloud." What specific CALIOP vfm and MODIS Cloud criteria are used to classify clouds into these categories for the DARES calculations?*

In Table 2, we've added:

"Table 3 lists which satellite-derived criteria are used to define four atmospheric scenarios"

In Table 3, CALIOP and MODIS are now replaced by CALIOP$_{VFM}$ and MODIS$_{Cloud}$

9. *DAREP Applicability: Section 2.2 emphasizes DAREP is specific to ORACLES conditions. For the broader goal of assessing DARE globally, are there plans to develop similar parameterizations for other regions/aerosol types, or will the full DARES framework always be the primary tool?*

We've added in section 2.2:

"We emphasize that this parametrization only represents the relationship between DARE and aerosol and cloud properties as sampled over the ORACLES study region and during the ORACLES timeframe. Outside of this framework (i.e., other regions of the globe and other seasons), different aerosol and cloud types can alter the DARE to cloud and aerosol relationship. To our knowledge, there are no current plans to extend the parameterization behind DARE_param

to other times and regions of the globe. Consequently, we will not be able to assess global DARE_obs results in future studies using DARE_param."

> 10. *Future Work - C3M Data (Section 4): The mention of C3M data (which relies on CloudSat) for future multi-cloud scenarios is relevant. While CloudSat's operational mode has changed and it's no longer in the A-Train, historical C3M data is extensive. For ongoing and future analyses, alternatives or updated multi-sensor products might be needed if relying on contemporaneous data with new missions like EarthCARE. This is more of a consideration than an error.*
>
> *And*
>
> 13. *Regarding the use of C3M data for multi-cloud scenarios: Given the changes in CloudSat's operational status, how does this impact the strategy for incorporating multi-layer cloud properties, especially for DARE calculations intended to span "multiple years" beyond the prime A-Train era? Will this rely more on the historical C3M dataset, or are there alternative/future multi-sensor cloud products (perhaps involving EarthCARE itself) that are being considered?*

We've modified this sentence in the discussion:

"We envision this additional scenario to use (i) the CALIPSO-CloudSat-CERES-MODIS (CCCM or C3M) (Kato et al., 2010, 2011) derived cloud heights and cloud microphysical properties or equivalent EarthCARE-derived product (e.g., as in Table 1 of Mason et al., (2024)) and (ii) MERRA-2 simulated aerosol extensive and intensive properties."

> 11. *The offset in DAREP vs. DARES is notably higher for 09/20/2016 (8.3 W·m⁻²) compared to other days. The paper suggests DAREP overestimation for high AOD on 09/18/2016. Does the larger offset on 09/20/2016 (which had the highest AODs) also primarily point to DAREP's single aerosol layer assumption or MERRA-2 SSA/ASY issues, or are there other potential contributors to this larger systematic difference on that specific day?*

We've added in section 3.3.1:

"When evaluating our semi-observational DARE_obs with coincident parametrized DARE_param over all types of clouds (i.e., S1, S2 and S3 in Table 3) and for our three case studies, we find a generally satisfying agreement ($R^2$=0.87 to 0.99, slope=0.80 to 0.99, offset =0.37 to 8.30, N=619 to 1067 in (1) Table 6). We posit that the slight differences between DARE_obs and DARE_param (see, for example, the mean cloudy DARE_param and DARE_obs values in panel (1) of Table 6) pertain to how they are computed. On the one hand, we assume MERRA-2's vertical distribution of SSA for the DARE_obs calculations, even though the SSA magnitude lies outside the observed SSA variability during ORACLES (i.e., as seen in Fig. 4b in Cochrane et al. (2021), the peak of the *in-situ* SSA values measured at 532 nm is between 0.85 and 0.86). By invoking this assumption, we can either overestimate DARE_obs if the MERRA-2 SSA value is too low or underestimate DARE_obs if the MERRA-2 SSA value is too high. For example, when computing DARE_theo (see Fig. A2), we record lower DARE_theo values (by ~10 W m⁻²) when adding more scattering aerosols (i.e., "continental") to already absorbing aerosols (i.e., "urban") over a thick cloud (COT=10). A second example is seen on 09/20/2016, where the two data points showing high AOD values above clouds (in yellow) and causing an offset in the DARE_param vs. DARE_obs regression line (~8 in Table 6) are likely due to an underestimation of MERRA-2 SSA, which in turn causes an overestimation of DARE_obs compared to DARE_param. On the other hand, while DARE_param is computed using the same AOD and cloud microphysical properties as DARE_obs, the DARE_param framework was developed specifically for aerosols above

homogeneous cloud conditions (i.e., S1) and thus might not apply as well to broken and/ or thin clouds (i.e., S2 and S3). The various amounts of S1, S2 and S3 cases during our three case studies (illustrated in Fig. 3) likely influence the DARE_param accuracy. We also note a distinctive feature in Fig. 6 on 09/18/2016 away from the 1:1 line for low AOD and CALIOP cloud fractions below 1 (black crosses). This feature is very likely due to cloud inhomogeneities paired with low AOD values."

> 12. *What are the anticipated major challenges in merging geostationary satellite data (which typically has coarser spatial resolution, different viewing geometries, and potentially different retrieval algorithms/sensitivities for aerosol and cloud properties) with the nadir-viewing, high-resolution Lidar/Imager data from A-Train/EarthCARE for consistent diurnal DARES calculations?*

We've added in the discussion:

"We note that aerosol and cloud retrievals from GEO satellites are in an earlier stage of development and less well-validated compared to their Low Earth Orbit (LEO) satellite counterparts. GEO aerosol and cloud retrievals are also currently often tied to specific GEO imagers and thus less global than their LEO counterparts. GEO AOD generally shows good agreement with ground-based AERONET AOD (e.g., low RMSE (0.12–0.17) in the case of the GEO Ocean Color Imager (GOCI) AOD over East Asia in Choi et al. (2019)) but have unique bias patterns related to the surface-reflectance assumptions in their retrieval algorithms (e.g., negative bias of 0.04 in GOCI AOD in Choi et al. (2019)). Recent improvements in algorithms consist in correcting surface reflectance, cloud masking and/ or fusing data from LEO and GEO imagers (e.g., Su et al. (2020), Zhang et al. (2020), Kim et al. (2020), and Choi et al. (2019)). In some cases, GEO AOD, although often biased, was shown to reproduce the AERONET AOD diurnal cycle (e.g., over Asia, on a daily average, GOCI AOD shows a diurnal variation of +20% to −30 % in inland sites according to Lennartson et al. (2018))."

Choi, M., Lim, H., Kim, J., Lee, S., Eck, T. F., Holben, B. N., Garay, M. J., Hyer, E. J., Saide, P. E., and Liu, H.: Validation, comparison, and integration of GOCI, AHI, MODIS, MISR, and VIIRS aerosol optical depth over East Asia during the 2016 KORUS-AQ campaign, Atmos. Meas. Tech., 12, 4619–4641, https://doi.org/10.5194/amt-12-4619-2019, 2019.

Kim, Jhoon, et al. "New era of air quality monitoring from space: Geostationary Environment Monitoring Spectrometer (GEMS)." *Bulletin of the American Meteorological Society* 101.1 (2020): E1-E22.

Lennartson, E. M., Wang, J., Gu, J., Castro Garcia, L., Ge, C., Gao, M., Choi, M., Saide, P. E., Carmichael, G. R., Kim, J., and Janz, S. J.: Diurnal variation of aerosol optical depth and PM$_{2.5}$ in South Korea: a synthesis from AERONET, satellite (GOCI), KORUS-AQ observation, and the WRF-Chem model, Atmos. Chem. Phys., 18, 15125–15144, https://doi.org/10.5194/acp-18-15125-2018, 2018.

Su, Tianning, et al. "Refining aerosol optical depth retrievals over land by constructing the relationship of spectral surface reflectances through deep learning: Application to Himawari-8." *Remote Sensing of Environment* 251 (2020): 112093.

Zhang, H., Kondragunta, S., Laszlo, I., and Zhou, M.: Improving GOES Advanced Baseline Imager (ABI) aerosol optical depth (AOD) retrievals using an empirical bias correction algorithm, Atmos. Meas. Tech., 13, 5955–5975, https://doi.org/10.5194/amt-13-5955-2020, 2020.